# Self-selection of food ingredients and agricultural by-products by the house cricket, *Acheta domesticus* (Orthoptera: Gryllidae): A holistic approach to develop optimized diets

**Juan A. Morales-Ramos**[1]*, **M. Guadalupe Rojas**[1], **Aaron T. Dossey**[2], **Mark Berhow**[3]

**1** USDA-ARS National Biological Control Laboratory, Stoneville, MS, United States of America, **2** All Things Bugs LLC, Oklahoma City, OK, United States of America, **3** USDA-ARS National Center for Agricultural Utilization Research, Peoria, IL, United States of America

\* juan.moralesramos@ars.usda.gov

**Data Availability Statement:** Data have been uploaded as Supporting Information files and to

## Abstract

The house cricket, *Acheta domesticus* L. (Orthoptera: Gryllidae) is one of the most important species of industrialized insects in the United States. Within the past five years the market of cricket powder as a food ingredient has been growing with increasing consumer interest on more sustainable sources of food. However, high labor costs of cricket production and high prices of cricket feed formulations result in cricket powder market prices much higher than other protein-rich food ingredients, making cricket powder only competitive within the novelty food market. In this study new diets formulated using by-products were developed using dietary self-selection followed by regression analysis. Crickets selected among seven different combinations of ingredients. Consumption ratios of food ingredients and by-products were used to determine macro and micro-nutrient intake. Regression analysis was used to determine the individual nutrient intake effect on cricket biomass production. Intake of vitamin C, sterol, manganese, and vitamins $B_1$ and $B_5$ had the most significant impact on live biomass production. Four diets were formulated based on this information and compared with a reference (Patton's 13) and a commercial diet. Although, crickets reared on Patton's diet 13 produced the most dry-weight biomass and developed the fastest, diet 4 (consisting of 92% by-products) generated the most profit (with a cost of $0.39 USD per kg) after an economic analysis that did not include the commercial formulation. Dry-weight biomass production was not significantly different among the four new diets and the commercial diet. This study demonstrated the value of dietary self-selection studies in developing oligidic insect diets and in studies of insect nutrition. This is the first such study involving farmed edible crickets and agricultural by-products. Four new cricket diet formulations contain between 62 and 92% agricultural by-products are included.

Research Gate with DOI numbers as follows: A-domesticus Self-Selection Ratios: DOI: 10.13140/RG.2.2.29623.47526 A-domesticus SS-Nutrient Intake Dry-Weight Basis: DOI: 10.13140/RG.2.2.36334.36161 A-domesticus Diet Evaluation Large Groups: DOI: 10.13140/RG.2.2.10749.10725 A-domesticus Diet Evaluation Small Groups: DOI: 10.13140/RG.2.2.17459.99363

**Funding:** One of the authors was affiliated to the company All Things Bugs, LLC, and this company was awarded the SBIR grant specified in the funding statement. This author's "Aaron T. Dossey" salary was paid by the SBIR grant. The company itself "All Things Bugs" did not provide any funding to the project. This project was executed under a USDA-ARS Cooperative Research and Development Agreement with All Things Bugs, LLC, which stated that all the results and any intellectual property obtained through this study belong to the United States Government. No company or funding organization participated in the planning of this study. Since in principle they are public property according to the policies of USDA-ARS on scientific research results obtained by government scientist as part of their duties. All the other three authors are US Government employees, who participated in this study as part of their duties.

**Competing interests:** I have read the journal's policy and the authors of this manuscript have the following competing interests: The author Aaron T. Dossey as the owner of All Things Bugs LLC is planning to commercialize cricket feeds formulated based on this study, however formulations for commercialization are different than the ones reported in this manuscript. Plans of All Things Bugs to commercialize diets using this study do not alter our adherence to PLOS ONE policies on sharing data and materials.

## Introduction

The house cricket, *Acheta domesticus* L. (Orthoptera: Gryllidae) is one of the most important species of industrialized insects in the United States [1]. It is estimated that the ten largest producers of house crickets in the United States collectively produce over 1,300 tones of live crickets per year [1]. Cricket production in the US is marketed mostly as pet food and fishing bait; however, within the past five years the market of cricket powder as a food ingredient has been growing, with 30–50 insect-based food companies forming in North America since 2013 alone (starting from less than 3 in 2012) [1].

Farmed insects are known to utilize less land, water, feed and other resources than traditional vertebrate livestock while generating lower levels of greenhouse gas emissions and contributing less to climate change [2, 3]. It has been estimated that insect production has a smaller environmental impact than chicken meat production using life cycle analysis [4]. Production of animal livestock uses approximately 70% of the land devoted to agriculture, or about 30% of the land on earth [5]. In addition, it is estimated that the US livestock population consumes more than seven times as much grain as consumed directly by the human population [6]. With the human population expected to rise to around 9.15 billion by 2050 [7, 8] and massive losses in biodiversity due to natural habitats being converted to agricultural land [8, 9], more sustainable sources of protein and other animal derived nutrients are urgently needed. However, the current costs of mass-producing crickets are still high, mostly due to primitive rearing practices that require too much labor [10]. As a result, market prices of cricket powder are much higher than other protein-rich food ingredients [1]. In addition to the high labor costs of cricket production, the high cost of cricket feed formulations adds to the final market price of cricket powder. Commercial feeds specially formulated for crickets are not common and tend to be relatively expensive (retail prices ranging between $7.00 to $28.00 USD per kg). Additionally, these feed formulations use ingredients such as ones derived from vertebrate animal livestock, which detract from the sustainability of insect farming via adding to the ecological footprint of those feeds. Many cricket producers in the US mix their own feed formulations, sometimes by modifying commercial chicken feeds based on published cricket diet studies [11, 12]. One way to reduce the costs of cricket production could be by reducing feed costs.

An ideal insect feed formulation from an environmental perspective would be made from plant based agricultural by-products or waste products which are not suitable for human consumption [4]. Some agricultural by-products may be adequate as ingredients for cricket feed formulations. Many agricultural by-products are considered waste because they are produced in much higher volumes than they can be utilized (S1 Text). Substituting food ingredients with agricultural by-products in cricket feed formulations may be a way to reduce costs. The use of agricultural by-products as insect diets has been explored for other species like *Tenebrio molitor* L., *Zophobas morio* Fab., and *Alphitobius diaperinus* Panzer (Coleoptera: Tenebrionidae) [13] and *Hermetia illucens* L. (Diptera: Stratiomydae) [14, 15], but no studies have been done on *A. domesticus*. Smetana et al. [4] determined that the environmental impact of insect production for feed and food can be greatly reduced by using agricultural by-products or waste products as insect feed (S1 Text).

By-products contain valuable nutrients that make them suitable for insect diets; however, developing viable diets for *A. domesticus* from by-products may require supplementation by the addition of other food ingredients. In addition, developing an oligidic (composed of ingredients that are not chemically defined) formulation using nutritionally complex ingredients may require years of experimentation and evaluation [16, 17]. Dietary self-selection by insects has been proposed as a holistic method for developing insect diets using complex ingredients

[16, 17, 18]. Dietary self-selection usually results in a mix ratio of ingredients being consumed by the insects that is optimal for development and reproduction because of their ability to regulate the intake of key nutrients by nutrient self-regulation [16]. Nutrient self-regulation abilities have been demonstrated in many species of insects, but is probably better developed in omnivorous species, which consume a wide variety of food types [16, 17]. No experimental proof has been published supporting the ability of *A. domesticus* to self-regulate their nutrient intake, but Patton [11] observed preferences in consumption of some diet ingredients when presented in pure form to groups of house crickets allowing them to choose. However, Patton [11] did not report using consumption ratios of individual ingredients as a criterion to develop diet formulations and his experiments were not designed with the idea of using self-selection as a tool for diet development.

Self-selection has been used to improve artificial diet formulations for the corn earworm, *Helicoverpa zea* (Boddie) (Lepidoptera: Noctuidae) [19]. Self-selection of three diet components by *Tribolium confusum* du Val (Coleoptera: Tenebrionidae) yielded a superior diet formulation compared with either of the ingredients alone [20]. Dietary supplement formulations for *T. molitor* were improved by using self-selection of different combinations of six ingredients to obtain optimal ratios for larval development [21]. Formulations mimicking the self-selected ratios not only shortened development time and increased larval survival and weight, but ratios self-selected by larvae also resulted in formulations that increased fecundity in adults [21]. There is not a report on the use of self-selection to develop diets for *A. domesticus*, but it is reasonable to assume that this method could be effective in this species based on the fact that *A. domesticus* is an omnivorous species and published observations [11] seem to indicate that they have the ability to self-select among dietary components. This compelled us to answer the question: can optimal diets be produced for house crickets based on their self-selection of complex ingredients, such as by-products? This depends on whether house crickets consume ingredients in such ratios that converge on consistent proportions of basic nutrients, such as lipid, protein, and carbohydrate. It also depends on whether the self-selected intake ratios of important nutrients can impact biomass production in a positive way. On this basis, the objectives of this study were 1) to test if *A. domesticus* selects among food ingredients and by-products for a diet that converges on consistent macro-nutrient ratios, which differ from those found in the individual ingredients; 2) to estimate self-selected intake ratios of key nutrients including lipid, protein, carbohydrate, sterols, vitamins, and minerals; 3) to determine if correlation exists between nutrient intake ratios and biomass productivity; 4) to formulate diets based on self-selected ratios of food ingredients and by products, and 5) to compare these diets with a commercial diet and a reference formulation on cricket development, survival, biomass production and food utilization efficiency.

## Materials and methods

This study consisted of two phases. In the first phase (phase 1), seven self-selection treatments were designed using different combinations of food ingredients and agricultural by-products. The food ingredients and by-products tested were selected on base of their current low price, high availability in the USA, moderate to high nutritional value, and historical use in insect diets or animal feed in general. Data obtained in phase 1 was used to estimate nutrient intake by self-selecting crickets and to determine optimal macro-nutrient intake ratios. In phase 2, four diet formulations were created based on results obtained in phase 1. The four diet formulations were evaluated by comparing biomass production and food utilization in large groups (estimated 1,000 crickets) and development time and survival in small groups (25 crickets).

## Rearing procedures

The stock colony used for this study originated from a single shipment of *A. domesticus* eggs donated by Millbrook Cricket Farms (Richland, Mississippi) received on August 28, 2015 showing excellent egg hatching conditions. Crickets were reared in unmodified 113.56 L (30 gal) plastic containers (74.3 L x 49.78 W x 43.43 cm H) (product No. 0218216, Bella Storage Solution, Leominster, MA, USA) without lid, filled with five standard cardboard 30-egg cartons (295 x 295 mm) cut in half and arranged in a horizontal stack at the bottom of the container. Water was provided by standard chicken water feeders (one per container) fitted with a polyurethane ring in the water tray to prevent the drowning of early instars. Food consisted of two commercial feed formulations: Purina Cricket Chow® (Purina Animal Nutrition LLC, Shoreview, MN, USA) and Coyote Creek organic cricket feed (Coyote Creek Organic Feed Mill, Elgin, TX, USA) mixed at a 2:1 ratio.

One g of first instar crickets (estimated to be 1500) were introduced to each container. Rearing containers were maintained in environmental chambers at 27 ± 1˚C, 65 ± 5% RH, and photoperiod of 12 hours photophase. Reproduction started eight weeks later by the introduction of oviposition devices into the containers, which consisted of polystyrene square boxes (110 L x 110 W x 35 mm H) (Product 156C, Pioneer Plastics Inc., North Dixon, KY, USA) filled with water-saturated coconut coir material (Nature's Footprint Inc., Bellingham, WA, USA) previously sterilized in an autoclave at 250˚C for 15 minutes. Adult crickets were provided with two oviposition devices, which were exposed for 2 days. Oviposition devices filled with eggs were placed inside polystyrene boxes (312 L x 230 W x 102 mm H) (Product 295C, Pioneer Plastics Inc., North Dixon, KY) and maintained at the same conditions described above until first instars eclosed 15 days later. Newly eclosed first instars were provided with nine crumbled pieces of tissue paper (Kimwipes®, Kimberly-Clark LLC, Roswell, GA, USA) as hiding places and with two petri dishes (10 x 35 mm diam.) filled with water-saturated poly-acrylamide crystals as a source of water.

## Phase 1

**Experimental units.**   Phase 1 experimental units consisted of groups of 3 g of first instar house crickets (approximately 4,500 crickets) reared in a 68.1 L (18 gal) plastic storage container (46.7 L x 59.4 W x 36.6 cm H) (Rubbermaid Inc., Huntersville, NC, USA). Each container was filled with 4 standard 30-egg cartons cut in half and stacked horizontally in the bottom of the container. Each experimental unit was also provided with a standard chicken water feeder as described above.

**Self-selection experiment.**   Newly eclosed first instars were randomly selected from the stock colony using a buccal aspirator fitted with a HEPA filter. First instars were weighed in groups of 500 mg using a precision balance (Mettler-Toledo AB104-S, Mettler-Toledo AG, Greifensee, Switzerland). Crickets were introduced to the experimental boxes until three grams of first instars have been added.

Seven self-selection treatments were designed using different combinations of food ingredients, algae, and by-products (Table 1). Treatment designations were NP1 and NP2 consisting mainly of 6 and 5 choices of food products, respectively; BP1, BP2, and BP3 consisting of 7, 4, and 8 choices of food and by products, respectively; and AL1 and AL2 consisting both of 6 choices of food, algae, and by products (Table 1). The food choices in each treatment were selected based on their ratios of macro-nutrient content [17] to make sure crickets had an adequate supply of macro-nutrients and diverse choices with different macro-nutrient ratios (Fig 1). Macro-nutrient ratios were calculated by dividing the content of each macro-nutrient (lipid, protein, and carbohydrate) by the sum of all macro-nutrient content in each food

**Table 1. Combination of food choices presented to cricket groups in portions of 5–10 g depending of cricket age, in seven self-selection treatments.**

| Food Ingredients | Treatment | | | | | | |
|---|---|---|---|---|---|---|---|
| | NP1 | NP2 | BP1 | BP2 | BP3 | AL1 | AL2 |
| Buckwheat Seed[1] | X | | X | X | X | | |
| Soy Flour (Full Fat)[1] | X | | | | | | |
| Soy Flour (Low Fat)[1] | | X | | | | | |
| Yellow Corn meal[1] | | X | | | | | |
| Sunflower Kernels[1] | | X | | | | | |
| Dry Milk (Whole)[2] | X | | | | | | |
| Dry Cabbage[3] | | | X | X | X | | |
| Lipid-Rich Algae[4] | | | | | | X | X |
| Protein-Rich Algae[4] | | | | | | X | |
| Spirulina[5] | | | | | | | X |
| Wheat Bran[1] | X | X | X | | | | |
| Rice Bran[6] | | | | | X | | X |
| Brewer's Yeast[7] | X | X | X | X | X | | |
| Alfalfa Pellets (Ground)[8] | X | | X | | X | | |
| Corn Dry Distillers Grain[9] | | | X | X | X | X | X |
| Canola Meal (Defatted)[10] | | | | | X | | X |
| Soybean Meal (Defatted)[11] | | | | | | X | |
| Rice Bran (Defatted)[6] | | | | | | X | |
| Peanut Hulls[12] | | X | | | | | |
| Soybean Hulls[11] | | | | | X | | X |
| Rice Hulls[6] | | | | | | X | |

[1] Bob's Red Mill (Milwaukie, OR, USA).

[2] Purchased in the local supermarket from brand names Nido Fortified (Nestle USA Inc., Glendale, CA, USA).

[3] Purchased fresh from the local supermarket and dried in a vacuum oven at 50°C and 180 mbar.

[4] AlgaVia (San Francisco, CA, USA).

[5] Earthrise (Irvine, CA, USA).

[6] Riceland Foods (Jonesboro, AR, USA).

[7] NOW (NOW Foods, Bloomingdale, IL, USA).

[8] Coyote Creek Organic Feed Mill (Elgin, TX, USA).

[9] Ergon Biofuels (Vicksburg, MS, USA) and Big River Resources Galva LLC (Galva, IL, USA).

[10] ADM Processing Co. (Chicago, IL, USA).

[11] Express Grain Oil Mill (Greenwood, MS, USA).

[12] Raw peanuts with shells were purchased locally with brand name Hines raw jumbo Virginia peanuts, Hines Nut Co. (Dallas, TX, USA). Shells alone were used in the experiments.

ingredient. For instance, protein ratio (Pr) was calculated as Pr = P / MN, where P is protein content, C is carbohydrate content, and L is lipid content in mg/100 g and MN is macro nutrient content = P+C+L in mg/100g. The ratios of carbohydrate (Cr) and lipid (Lr) were calculated in the same manner and the sum of Pr + C r+ Lr = 1 [17]. The nutrient content information of food products and spirulina algae was obtained from the USDA nutrient database [22], for the by-products this information was obtained from several published sources [23, 24, 25, 26, 27, 28, 29, 30, 31, 32, 33, 34, 35, 36]. In addition, information provided by the distributors that donated their products for this study (Ergon Biofuels, Big River Resources Galva LLC, Riceland Foods Inc., Express Grain Terminals LLC, and ADM Processing Co.) was cross referenced with published information on by-product nutrient content.

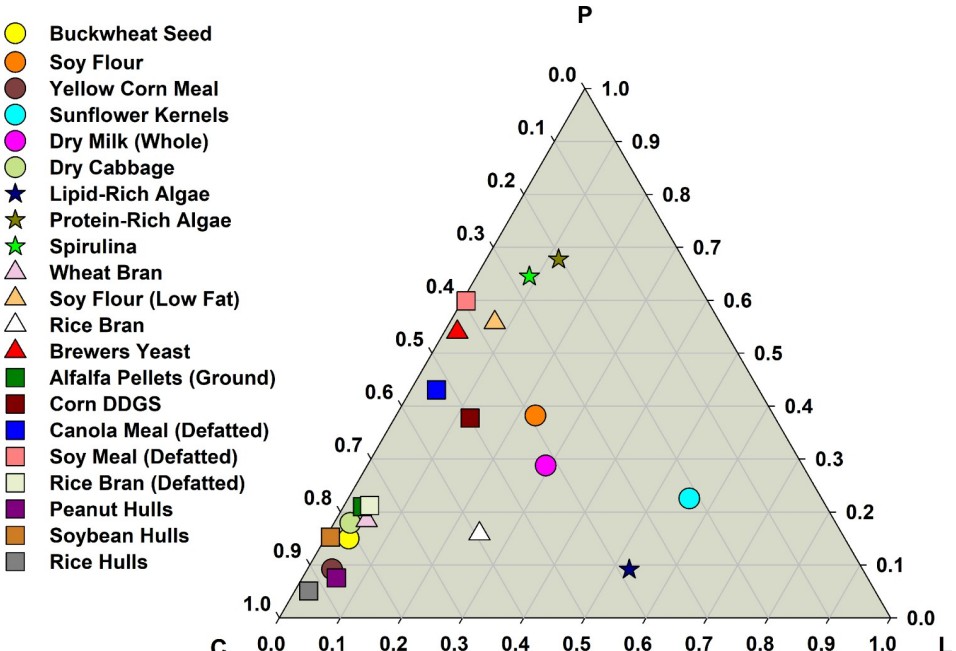

**Fig 1. Macro nutrient ratios of the ingredients used in seven self-selection treatments.** P = protein, L = lipid, C = carbohydrate. Ingredients with circle symbols are food grade ingredients; star symbols represent algae products; triangles represent by-products used as food; and squares represent by-products no used as food (but used to feed food producing animals).

Each treatment was replicated ten times (10 experimental units per treatment). Choices of four to eight food ingredients (depending on the treatment) were presented to the crickets in Petri dishes arranged radially in a paper plate positioned on top of the carton rearing substrate (S1 Fig). Experimental boxes were maintained at 27 ± 0.5°C, 65 ± 5% RH, and 14 h photophase for a period of eight weeks. Equal amounts consisting of 5 g of each food ingredient were provided at the beginning of the experiment. Each ingredient was replenished as cricket consumed them and a record was maintained of the amounts added of each ingredient to each of the experimental boxes of each treatment.

The experiment lasted eight weeks, which is an adequate production cycle period at 27°C [37]. At the end of this period the uneaten remining portions of each food ingredient were collected and labelled with ingredient name, treatment, repetition number, and date. The water feeders and rearing substrate were removed from the boxes. The cricket frass from each box was collected in a Petry dish and labelled with treatment, repetition number, and date. Remaining food and frass were dried in a vacuum (oven at 50°C and 180 mbar) for a period of 48 hours. Crickets were separated from the rearing substrate, placed in a pre-weighed plastic container, and weighed live in group to determine the ending live biomass (g) of each experimental unit.

**Data analysis.** The consumption of each ingredient in mg ($I_i$) by the crickets was calculated as total dry-weight added (mg) of ingredient 'i' minus dry-weight remaining (mg) of ingredient 'i' in each of the seven treatments. Proportions consumed of each ingredient ($PI_i$) from the total food consumed (FC) were calculated as food ingredient consumption divided by total food consumed ($PI_i = I_i/FC$) in each of the seven treatments, where FC = $\Sigma\, l_i$. Food assimilated (FA) was calculated as FC–Frass and food assimilation ratio was calculated as FA/FC.

The intake of macro and micro-nutrients in each treatment was calculated using the nutrient matrix as described by Morales-Ramos et al. (2014) [17]. Macro-nutrient ratio intake was calculated as explained in the self-selection experiment subsection using the data obtained from the nutrient matrix operations for the intake of protein, lipid and carbohydrate. The nutrient matrix was also used to estimate intake of micro-nutrients including vitamins: A, E, C, B1, B2, B3, B4, B5, B6, B9, and K; minerals: Ca, Mg, Fe, K, Na, P, Zn, Cu, Mn, and Se; and sterol (phytosterols + ergosterol + cholesterol). Content of all nutrients was expressed in mg per g, except for vitamin A, which was expressed as international units (IU).

Data consisting of ending biomass and total food consumed were analysed using general linear mixed model (GLMM) and least square means from different treatments were compared using the Tukey-Kramer HSD test at α = 0.05. The mixed model GLMM has the capability of analysing random effects of different distributions including normal, binomial, and Poisson unlike conventional GLM, which is restricted to normaly distributed random effects [38]. Generalized linear mixed model GLMM was also used to analyse and compare ingested macro-nutrient ratios among treatments. Because GLMM supports binomial distribution, ratios did not require to be root square arcsine converted to eliminate the bias introduced by proportional values [39]. This method was also used to analyse and compare food assimilation ratios among treatments.

Simple linear regression analysis was used to determine the impact of total food consumption on ending cricket biomass across treatments. The impact of consumption of each individual ingredient on ending cricket biomass was assessed using simple linear regression within treatments and across treatments using the same ingredient. Multiple linear regression was used to determine the impact of multiple nutrients (macro and micro) on ending cricket biomass across treatments. A regression model with optimal number of parameters was determined using stepwise regression followed by backward elimination techniques [40, 41]. Then the significance of these parameters was assessed using multiple regression and response surface analyses. Significant quadratic effects and interactions were included in the final model by backwards elimination of the response surface model. Statistical software used for all the analyses was JMP ver. 12 [42].

### Phase 2

**Diet formulations.** Four diet formulations were produced using mostly by-products based on results from phase 1 (Table 2). Diet 1 was formulated using the consumption ratios obtained in phase 1 treatment BP1, which produced the second highest mean cricket biomass. Diet 2 was formulated as a modification of consumption ratios from treatment BP2 and the addition of 3 more ingredients while maintaining the macro-nutrient ratios consistent with the observed in the experiment. Diet 3 was formulated using the mean consumption ratios observed in treatment BP3, which produced the highest cricket biomass, adjusted after the elimination of soybean hulls from the formula to maintain the same macro-nutrient ratios. Diet 4 was formulated using by-product ingredients that were highly consumed in all treatments of phase 1 and balanced to maintain the macro-nutrient ratios observed in treatment BP3 (Table 2).

The new diet formulas were compared with reference cricket diet number 13 developed by Patton (1967) [11]. Although diet 13 was not the best performer in the study conducted by Patton (1967) [11] (diets 16 and 3 performed better), this formulation was the third best and resembled the closest the self-selected macro nutrient ratios observed in this study. This made it a better reference diet for comparison with the new formulations. Patton's diet 13 was modified to conform with ingredients currently available and nutritionally consistent. Skim milk

**Table 2. Experimental diet formulations and control reference diet in grams per 100 grams and estimated macro nutrient ratios.**

| Food Ingredients | Diets | | | | |
|---|---|---|---|---|---|
| | 1 | 2 | 3 | 4 | Patton 13 |
| Buckwheat Seed | 29 | 25 | 19 | – | – |
| Soy Flour (Low Fat) | – | – | – | – | 10 |
| Yellow Cornmeal | – | – | – | – | 35 |
| Dry Milk (Whole) | – | – | – | – | 7.5 |
| Dry Milk (Skim) | – | – | – | – | 7.5 |
| Dry Beef Liver Defatted | – | – | – | – | 5 |
| Dry Cabbage | 9 | – | 10 | 8 | – |
| Wheat Bran | 8 | 5 | – | – | 30 |
| Rice bran | – | 5 | 17 | 20 | – |
| Brewer's Yeast | 16 | 10 | 17 | 8 | 5 |
| Alfalfa Pellets (Ground) | 4 | 12 | 3 | 4 | – |
| Corn DDGS | 34 | 38 | 28 | 30 | – |
| Canola Meal (Defatted) | – | 5 | 6 | 10 | – |
| Rice Bran (Defatted) | – | – | – | 20 | – |
| Estimated Major Nutrient Ratios | | | | | |
| Lipid | 0.06 | 0.075 | 0.091 | 0.105 | 0.062 |
| Protein | 0.3 | 0.297 | 0.308 | 0.3 | 0.268 |
| Carbohydrate | 0.64 | 0.628 | 0.601 | 0.595 | 0.67 |

was replaced by a mix 1:1 of defatted dry milk and whole dry milk; wheat middling was replaced by wheat bran; and dry defatted pork liver was replaced by dry defatted beef liver. The remaining ingredients (corn meal, soy flour, and brewer's yeast) were as reported by Patton (1967) [11]. Diets were prepared by mixing all ingredients in the proportions presented in Table 2 using a high-speed electric mixer (NutriBullet, Model NB-201, Homeland Housewares, LLC, Los Angeles, CA, USA) for a period of 30 seconds. Mixed diets were presented to the crickets dry in powder form in Petri dishes of three different sizes depending of cricket age: 1) from start to 2 weeks 5g of diet in dishes 12 mm H x 55 mm diam., 2) from 2 to 4 weeks 10 g of diets in dishes 20 mm H x 60 mm diam., and 3) from 4 to 7 weeks 20 g in dishes 25 mm H x 90 mm diam.

All diet formulas were also compared with a commercial cricket diet (Purina Cricket Chow, Purina Animal Nutrition LLC, Shoreview, MN, USA). Formula for this diet is not available, but ingredients reported by the manufacturer include: Ground corn, wheat middlings, ground soybean hulls, dehulled soybean meal, porcine meat meal, porcine animal fat preserved with BHA, cane molasses, fish meal (menhaden), salt, calcium carbonate, di-methionine, magnesium oxide, choline chloride, manganous oxide, zinc oxide, ferrous carbonate, niacin, copper sulfate, calcium pantothenate, (di-alpha tocopheryl acetate, riboflavin, thiamin mononitrate, vitamin A acetate, zinc sulfate, folic acid, menadione sodium bisulfite complex, calcium iodate, pyridoxine hydrochloride, sodium selenite, cobalt carbonate, cholecalciferol, and vitamin B12 at undisclosed ratios. The commercial diet was powdered using the same high-speed mixer described above to eliminate particle size as a factor.

**Large group experiment.** The objective of this experiment was to compare the effects of diet treatment on cricket growth, biomass production, and food utilization under conditions that simulate a large-scale production system. Experimental units consisted of groups of 660 mg of first instars (estimated 1,000 crickets) reared in plastic storage boxes with dimensions of 40.1 x 67.8 x 33.5 cm (62 L) (Rubbermaid Inc., Huntsville, NC, USA). These experimental

containers were narrower and longer than the containers used in phase 1 experiments. The size of the containers was chosen to be able to fit six containers (a full 6-treatment repetition) inside each environmental chamber. Rearing substrate consisting of 4 egg cartons cut in half and stacked horizontally as in phase 1 experiments. Conventional chicken water feeders were not used in this experiment because a substantial amount of cricket frass could not be recovered for measurement. Instead, inverted water feeders designed to allow crickets to drink upside-down were used in these experiments. Inverted water feeders consisted of a deposit with a screened bottom filled with saturated polyacrylamide crystals and suspended with four 1.5-cm legs. Crickets sucked water from the polyacrylamide crystals through the screen while standing inverted (US utility patent application No. 15935403).

One of six diet choices (Table 2 diets or commercial diet) was provided to treatment groups consisting of 15 experimental units. Each experimental unit was provided initially with five grams of diet. The diet was replenished as it was consumed by the crickets and the amount of diet added to each experimental box of each treatment was recorded. The water content of each diet formula was determined by measuring the weight loss of ten 500-mg samples of each formula after drying them in a vacuum oven at 50˚C and 80 hPa pressure for 48h. The information on water content of each formula was used to estimate the dry weight of the total amount of food added to each experimental unit and treatment. This method eliminated potential errors originating from potential effects of the drying process on the nutritional integrity of the diet formulas.

Experimental units were maintained in environmental chambers (Percival) at 27˚C, 65% RH, and 12:12 h (L:D) photoperiod. A full repetition (6-treatments) was maintained in each chamber and the position of the experimental units was daily rotated counterclockwise to eliminate any position effects inside the chambers. Experimental units were maintained at these conditions for a period of seven weeks (49 d). At the end of the experimental period, crickets from each treatment group were removed from their box, weighed alive and recorded. Cricket groups were labeled with their respective treatment and repetition number and frozen at -28˚. After frozen, crickets from each experimental box were counted and the number of adults, last instars, and younger nymphs were recorded. The group weight of adults, last instars and younger nymphs was determined and recorded. Frozen cricket groups (adults, last instars, and young nymphs separately) were dried in a vacuum oven at the conditions described above for a period of 72h and weighed to determine the ending dry-weight biomass for each experimental unit and recorded with their respective treatment and repetition number information. The frass was carefully collected from each experimental box as well as the remaining uneaten food, labeled with the respective treatment and repetition number, and dried in a vacuum oven at the same conditions described above.

**Food utilization.** Live weight gain (LWG) was determined by subtracting the initial nymphal weight (660 mg) from the ending weight of live crickets from each box. Dry weight gained (DWG) was calculated the same way using dry-weight biomass data. The initial cricket dry weight in each experimental box could not be determined by direct measurement, so it was estimated based on the water content of early instars *A. domesticus* reported by Finke (2002) [43] to be 141.9 mg for every 660 mg of live weight. Dry-weight food consumed (FC) was calculated by subtracting the dry weight of the remaining food from the estimated dry weight of the total food consumed in each experimental box. Dry-weight food consumed and dry-weight biomass gained data were used to calculate food utilization parameters as described by Waldbauer (1968) [44]. Food assimilated (FA) was calculated by subtracting the frass dry-weight from the dry-weight food consumed (FA = FC–frass). Efficiency of conversion of ingested food (ECI) was calculated as ECI = DWG * 100 / FC and efficiency of conversion of assimilated food (ECA) was calculated as ECD = DWG * 100 / FA [44].

**Small group experiment.** The objective of the small group experiment was to determine the impact of diet on development time, survival, and individual weight of newly emerged adult crickets. Experimental units consisted of groups of 25 crickets confined in cages constructed from polystyrene boxes (19.5 L x 14 W x 9.5 cm H) (product 079C, Pioneer Plastics, Inc, Dixon, KY, USA) modified with the addition of 6 screened windows (2.8 cm diam.) on the sides (one per smaller side and 2 per longer side) and two screened windows (6.5 cm diam.) on the cover. Each box was lined with a paper napkin at the bottom to facilitate cricket movement and a piece of egg carton material (9.5 x 9.5 cm) was added to provide hiding space. Water was provided by inverted water dispensers constructed from petri dishes (2 cm high x 6 cm diam.) (Nunc 4036, Nalgene Nunc International, Rochester, NY) with screened bottoms (No. 20, 850 μm openings) and filled with saturated polyacrylamide. Food was provided (500 mg) in small petri dish bottoms (1 cm high x 3.5 cm diam.) placed next to the water dispensers.

Newly eclosed first instars were randomly selected and counted by using a buccal aspirator fitted with a HEPA filter. The same diet treatments tested in the large group experiments were tested in the small group experiment (Table 2). Eight experimental units per treatment were placed inside an environmental chamber having 8 shelves in each of two sides and environmental settings of 27˚C, 65% RH, and 12:12 h (L:D) photoperiod. One experimental unit of each of the six treatment was placed in every shelf (6 per shelf) and distributed using a random square design. Water and food were replenished as consumed and the weight of the food added to each experimental unit was recorded.

After six weeks, experimental units were monitored daily for the presence of adult crickets. When present, adult crickets were sexed, weighed, and frozen at -25˚C. The live weight and sex of each cricket was recorded along with its corresponding experimental unit number, diet treatment and date of emergence. Frozen crickets were dried in a vacuum oven at 50˚C and 80 hPa pressure for 48 h and weighed. The dry weight of each cricket was recorded along with its corresponding experimental unit number, diet treatment and date of emergence. This procedure continued until all the crickets in all experimental units had completed development. The development time of each cricket was determined by the number of days between their date of first instar eclosion and the date of adult emergence.

**Data analysis.** Data obtained from the large group experiment consisting of dry-weight food consumed (g), ending cricket live biomass (g), dry-weight biomass gain (g), ending number of crickets, and mean individual adult and last instar weights (mg) (live and dry-weights) were analyzed for the effect of different diets using GLMM (generalized linear mixed model) and least square means were compared among diet treatments using the Tukey-Kramer HSD test for least square means at $\alpha = 0.05$. The same analysis procedure was used to compare data expressed as proportions including proportion of adults, dry-weigh proportion of adults and last instars, food conversion proportion, ECI, and ECA expressed as proportions. Models for the large group experiments were constructed as: dependent variable (food consumption, food assimilation, live weight biomass, and ECI) = treatment variable (diet = nominal) + effects (random or binomial). A second set of models were constructed as: dependent variable (food consumption, food assimilation, live weight biomass, and ECI) = treatment variable (diet = nominal) + % adults + effects (random or binomial) to determine the impact of adult percentage on dependent variables

Data obtained from the small group experiment consisting of development time in days and adult weight of females and males was analysed using the same procedure as above consisting of GLMM and Tukey-Kramer HSD test to compare least square means. Cricket survival from first instar to adult was analysed using contingency analysis and survival among diet treatments was compared using analysis of means (ANOM) for proportional data [42, 45]. The

procedure ANOM generates an overall mean proportion and two decision levels, upper and lower (UDL and LDL). Proportions outside the decision levels are significantly higher (value above UDL) or lower (value below LDL) than expected [45]. Models for the small group experiment were constructed as: dependent variable (live weight biomass gain, dry weight biomass gain, development time) = treatment variable (diet = nominal) + sex (Boolean) + tray position (nominal) + effects (random or binomial).

**Economic analysis.** Revenues per kg of cricket powder produced were calculated for diets 1 to 4 and Patton's diet 13. The commercial diet was not included in the economic analysis comparisons because its market price includes costs of milling, mixing and packing as well as ingredient shipment to the milling site. All these costs were not considered in economic analysis of the other five diets. Kilograms of food consumed per kg of dry-weigh cricket biomass production was calculated as FC / DWG using results data from the large group experiment. Cost of the diets per kg was calculated based on current pricing of diet ingredients from Table 2 obtained from internet resources [46, 47, 48]. The current price per kg of cricket powder was obtained from internet sources of 9 different companies and an average was calculated after converting pricing and weight units to USD per kg. The average cost of cricket powder per kg was $93.05 USD ranging from $77.16 to $123.37 as on May 2019. Revenue per kg of cricket powder, not including diet mixing costs and rearing labour, was calculated as the price of cricket powder per kg minus the cost of diet consumed per kg of dry-weight cricket biomass produced.

Revenues from cricket powder production per unit of rearing space were also compared among diets 1 to 4 and Patton's diet 13. Lundy and Parrella (2015) [49] estimated that 96 standard egg carton pieces (30 x 30 cm) provided an area of approximately 172, 800 cm$^2$. Based on this estimate, the area of one egg carton provides an approximate area of 1,800 cm$^2$ of rearing space. Based on the size of egg cartoons and the vertical space they take when stacked (5 cm), it is estimated that a stack of 211.89 egg cartons will take a space of approximately one m$^3$ and provide 381,398 cm$^2$ of effective rearing area. Data of dry-weight cricket biomass production and food consumed per cycle was obtained from results of the large group experiment. The means of dry weight cricket biomass and food consumption obtained for each diet in the large group experiment were transformed to a per m$^3$ basis by multiplication with a conversion coefficient (CC) calculated as rearing space per m$^3$ divided by rearing space per experimental unit. Revenue per m$^3$ per production cycle was calculated as (DWG * $93.05 –FC * Cost per kg) * CC. The revenue per year was calculated assuming a 10-wk production cycle consisting of 2 weeks for oviposition and egg development plus 8 weeks of development to reproductive adults, which results in five production cycles per year approximately.

## Results

### Phase 1

Mean consumption data of each ingredient by treatment are presented in Table 3. Total food consumed was significantly different among treatments ($F = 28.76$; df 6, 59; $P < 0.0001$) (Fig 2A). Crickets in treatment BP3 consumed significantly more food than all the other treatments. Overall, food consumption had a significant positive impact on ending live biomass ($R^2 = 0.865$; $F = 410.35$; df 1, 64; $P < 0.0001$) and this translated into significant differences in ending live biomass among treatments ($F = 22.21$; df 6, 59; $P < 0.0001$). Treatment BP3 produced significantly more live biomass per experimental unit than the rest of the treatments except for treatment BP1 (Fig 2B).

There was significant difference in the percentage ratio at which each food ingredient was consumed within treatments: NP1 ($F = 166.2$; df 5, 42; $P < 0.0001$), NP2 ($F = 96.9$; df 4, 35;

**Table 3. Mean consumption (g) per box of individual food ingredients in seven self-selection treatments.**

| Food Ingredient | Treatment | | | | | | |
|---|---|---|---|---|---|---|---|
| | NP1 | NP2 | BP1 | BP2 | BP3 | AL1 | AL2 |
| Buckwheat Seed | 17.9 ± 7.7 | - | 83.5 ± 27.5 | 58.8 ± 21.5 | 86.5 ± 24.7 | - | - |
| Soy Flour Full Fat | 3.6 ± 1.7 | - | - | - | - | - | - |
| Soy Flour Low Fat | - | 14.3 ± 2.8 | - | - | - | - | - |
| Yellow Corn Meal | - | 10.3 ± 2.7 | - | - | - | - | - |
| Sunflower Kernels | - | 11.4 ± 2.1 | - | - | - | - | - |
| Whole Dry Milk | 28.7 ± 13.4 | - | - | - | - | - | - |
| Dry Cabbage | - | - | 27.7 ± 12.5 | 29.1 ± 10.7 | 46.2 ± 16.5 | - | - |
| Lipid Rich Algae | - | - | - | - | - | 11.5 ± 3.5 | 25.5 ± 3.1 |
| Protein Rich Algae | - | - | - | - | - | 6.5 ± 3.5 | - |
| Spirulina | - | - | - | - | - | - | 34.1 ± 4.6 |
| Wheat Bran | 22.4 ± 7.5 | 30.4 ± 12.4 | 25.1 ± 13.2 | - | - | - | - |
| Rice Bran Whole | - | - | - | - | 76.8 ± 21.0 | - | 54.5 ± 7.2 |
| Brewer's Yeast | 14.1 ± 5.9 | 15.3 ± 4.2 | 45.5 ± 13.5 | 43.0 ± 16.1 | 80.2 ± 19.9 | - | - |
| Alfalfa Pellets | 30.0 ± 14.1 | - | 11.6 ± 3.8 | - | 13.2 ± 5.1 | - | - |
| Corn DDGS | - | - | 98.0 ± 32.1 | 82.2 ± 24.9 | 130.8 ± 37.7 | 34.1 ± 9.8 | 51.9 ± 9.9 |
| Canola Meal | - | - | - | - | 31.6 ± 13.0 | - | 28.5 ± 6.2 |
| Soy Meal Defatted | - | - | - | - | - | 12.4 ± 4.5 | - |
| Rice Bran Defatted | - | - | - | - | - | 66.4 ± 18.2 | - |
| Peanut Shells | - | - | 1.1 ± 0.7 | - | - | - | - |
| Soybean Hulls | - | - | - | - | 1.9 ± 1.2 | - | 4.7 ± 0.7 |
| Rice Hulls | - | - | - | - | - | 0.2 ± 0.1 | - |

Mean ± standard deviation

$P < 0.0001$), BP1 ($F = 453.8$; df 6, 63; $P < 0.0001$), BP2 ($F = 314.3$; df 3, 36; $P < 0.0001$), BP3 ($F = 826.2$; df 7, 72; $P < 0.0001$), AL1 ($F = 538.8$; df 5, 54; $P < 0.0001$), and AL2 ($F = 663.7$; df 5, 54; $P < 0.0001$). Crickets showed preference for some ingredients, which were consumed at significantly higher ratios within each treatment (Table 4). The consumption preference of some ingredients differed among treatments and seemed to be affected by the food choices available. For instance, ground alfalfa pellets was the most preferred ingredient in treatment NP1, but was one of the least preferred in treatments BP1 and BP3; wheat bran was significantly more preferred than brewer's yeast in treatment NP2, but the opposite was true for treatment BP1; and corn DDGS was significantly more preferred than rice bran in treatment BP3, but treatment AL2 showed the opposite (Table 4).

The macro-nutrient intake ratios were highly consistent within treatments (low standard deviation values) but differed significantly between treatments: lipid ratio ($F = 1042$; df 6, 59; $P < 0.0001$), protein ratio ($F = 53.3$; df 6, 59; $P < 0.0001$), and carbohydrate ratio ($F = 253.1$; df 6, 59; $P < 0.0001$) (Table 5). The highest lipid intake ratios were observed in treatments AL2 (0.195 ± 0.005) (mean ± standard deviation) and NP2 (0.122 ± 0.009), which also showed the lowest carbohydrate intake ratios (0.476 ± 0.003 and 0.558 ± 0.02, respectively). Conversely, treatment BP1 showed the lowest lipid intake ratio (0.059 ± 0.001) and the highest carbohydrate intake ratio (0.645 ± 0.006) (Table 5). Treatments AL2, BP2, and NP2 showed the highest protein intake ratios (0.329 ± 0.005, 0.321 ± 0.005, and 0.32 ± 0.011, respectively) and the lowest protein intake ratio was observed in treatment NP1 (0.265 ± 0.008) (Table 5). The overall means of macro nutrient intake ratios across treatments were 0.106 ± 0.044, 0.307 ± 0.021, and 0.587 ± 0.055 for lipid, protein, and carbohydrate intake, respectively. Statistical distribution of

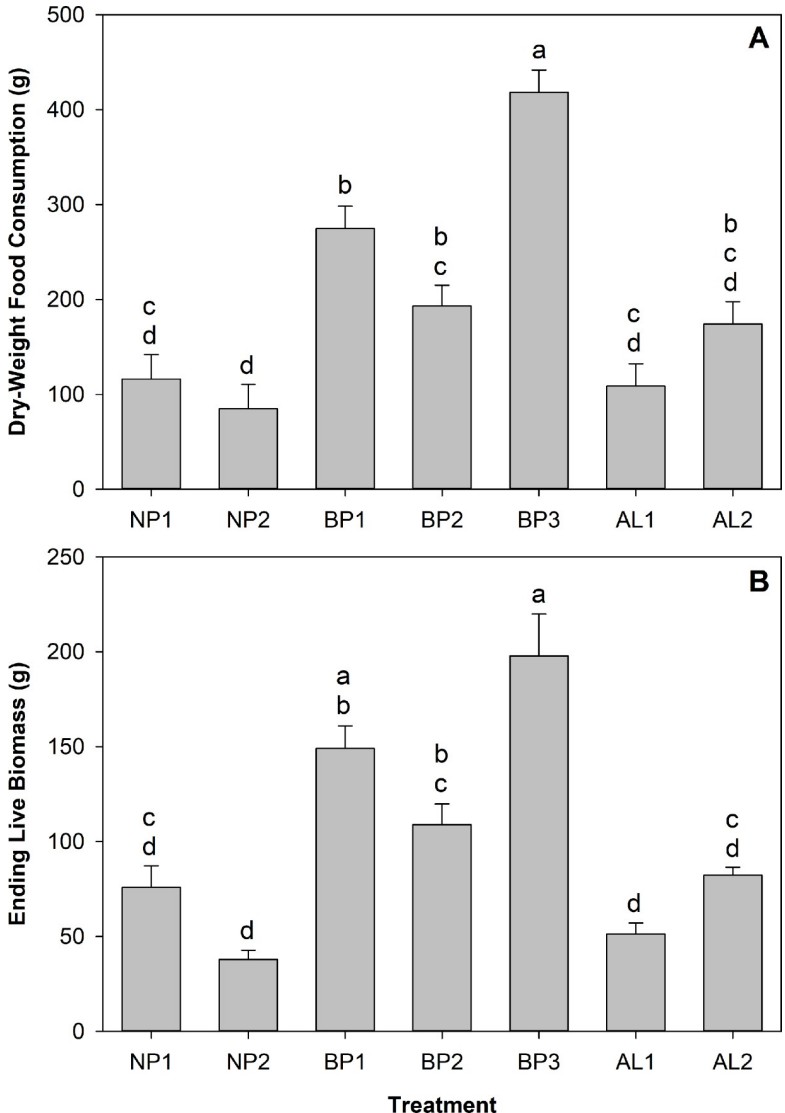

**Fig 2.** Means of dry-weight total food consumed (A) and live biomass gain (B) by cricket groups in seven self-selection treatments. Brackets represent standard deviation. Columns with the same letter are not significantly different after Tukey-Kramer HSD test at α = 0.05.

the intake ratios of lipid and carbohydrate showed high frequencies of extreme values for treatment AL2, which grouped within the 90–100 percentiles for lipid intake and within the 0–10 percentiles for carbohydrate intake (Fig 3). Eliminating treatment AL2 from the overall means, the macro nutrient intake ratios were 0.09 ± 0.025, 0.3 ± 0.02, and 0.61 ± 0.03 for lipid, protein, and carbohydrate intake, respectively.

The intake of nutrients by crickets in each of the treatments, as estimated by the nutrient matrix operation, are presented in Table 6. Linear regression analyses of single nutrients versus ending live biomass showed that neutral detergent fiber, sterol, carbohydrate, and vitamins C, A, K, B2, and B4 had a significant positive impact on ending live biomass. In contrast, lipid, vitamin E, calcium, iron, magnesium, zinc and manganese had a significant negative effect on ending live biomass. However, when considered together in a multiple regression model, nutrients affected the ending live biomass differently.

**Table 4. Mean consumption percentages of individual food ingredients by cricket groups in seven self-selection treatments.**

| Food Ingredient | Treatment | | | | | | |
|---|---|---|---|---|---|---|---|
| | NP1 | NP2 | BP1 | BP2 | BP3 | AL1 | AL2 |
| Buckwheat Seed | 15.4 ± 3.3c | - | 28.6 ± 1.8b | 27.3 ± 2.3b | 18.5 ± 0.9b | - | - |
| Soy Flour Full Fat | 3.1 ± 0.6e | - | - | - | - | - | - |
| Soy Flour Low Fat | - | 18.0 ± 3.0b | - | - | - | - | - |
| Yellow Corn Meal | - | 12.7 ± 0.5c | - | - | - | - | - |
| Sunflower Kernels | - | 14.3 ± 1.8c | - | - | - | - | - |
| Whole Dry Milk | 24.2 ± 1.8a | - | - | - | - | - | - |
| Dry Cabbage | - | - | 9.1 ± 2.4d | 13.6 ± 1.6d | 9.8 ± 1.2d | - | - |
| Lipid Rich Algae | - | - | - | - | - | 8.7 ± 0.8c | 12.8 ± 0.7c |
| Protein Rich Algae | - | - | - | - | - | 4.8 ± 2.0d | - |
| Spirulina | - | - | - | - | - | - | 17.2 ± 0.9b |
| Wheat Bran | 19.7 ± 3.0b | 36.1 ± 4.9a | 8.2 ± 1.6d | - | - | - | - |
| Rice Bran Whole | - | - | - | - | 16.5 ± 1.0c | - | 27.5 ± 2.0a |
| Brewer's Yeast | 12.1 ± 2.1d | 18.8 ± 1.0b | 15.8 ± 1.2c | 19.9 ± 1.1c | 17.5 ± 1.7bc | - | - |
| Alfalfa Pellets | 25.5 ± 3.3a | - | 4.3 ± 2.1e | - | 2.8 ± 0.5f | - | - |
| Corn DDGS | - | - | 33.6 ± 1.4a | 39.2 ± 2.5a | 28.0 ± 0.8a | 26.2 ± 3.1b | 25.9 ± 1.7a |
| Canola Meal | - | - | - | - | 6.5 ± 1.4e | - | 14.2 ± 1.6c |
| Soy Meal Defatted | - | - | - | - | - | 9.3 ± 2.3c | - |
| Rice Bran Defatted | - | - | - | - | - | 50.8 ± 4.9a | - |
| Peanut Shells | - | - | 0.4 ± 0.1f | - | - | - | - |
| Soybean Hulls | - | - | - | - | 0.4 ± 0.2g | - | 2.4 ± 0.4d |
| Rice Hulls | - | - | - | - | - | 0.4 ± 0.2e | - |

Mean ± standard deviation; means with the same letter within columns are not significantly different after Tukey-Kramer HSD test at α = 0.05.

The stepwise procedure of a full model including all the nutrients as independent variables determined that a model including only six variables (seven parameters) explained the dependent variable the best (Table 7). The model included sterol, manganese, and vitamin C with positive effects and protein, and vitamins B1, and B5 with negative effects on the dependent variable, ending live biomass (Table 8). Response surface analysis of the model followed by backwards elimination of non-significant variables yielded a nine-parameter model, which included quadratic effects of vitamin B5 and interactions of manganese with vitamins B5 and sterol with vitamin B1 (Table 9). These analyses also eliminated protein from the model, which no longer had significant impact on the dependent variable. It appears that vitamins B1 and B5

**Table 5. Macro nutrient intake ratios in seven self-selection treatments with different combinations of food ingredients.**

| Treatment | Lipid | Protein | Carbohydrate |
|---|---|---|---|
| NP1 | 0.106 ± 0.004d | 0.265 ± 0.008e | 0.629 ± 0.011b |
| NP2 | 0.122 ± 0.009b | 0.32 ± 0.011ab | 0.558 ± 0.02e |
| BP1 | 0.059 ± 0.001f | 0.296 ± 0.005d | 0.645 ± 0.005a |
| BP2 | 0.061 ± 0.002f | 0.321 ± 0.005a | 0.617 ± 0.006bc |
| BP3 | 0.089 ± 0.003e | 0.308 ± 0.005bc | 0.603 ± 0.005c |
| AL1 | 0.114 ± 0.005c | 0.305 ± 0.016cd | 0.582 ± 0.018d |
| AL2 | 0.195 ± 0.005a | 0.329 ± 0.005a | 0.476 ± 0.003f |

Mean ± standard deviation. Means with the same letter (within columns) are not significantly different after Tukey-Kramer HSD test at α = 0.05.

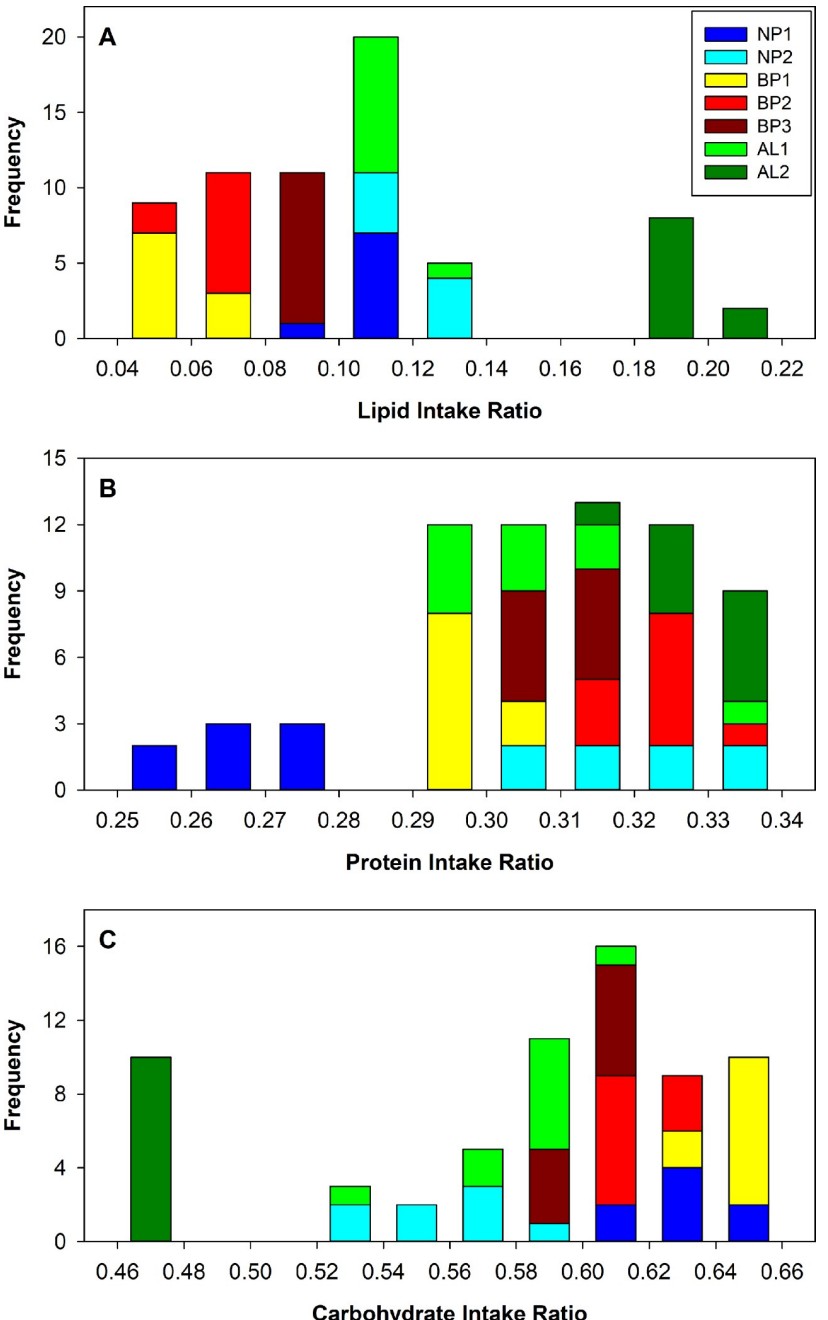

**Fig 3.** Combined histograms of intake ratios of lipid (A), protein (B), and carbohydrate (C) of house crickets in seven self-selection treatments represented by different color bands within bars.

in addition of having negative effects on their own, also interact negatively with other variables resulting in an overall negative effect on the dependent variable (Table 9). A solution of this model for an ending live biomass of 200 g per experimental unit, yielded 310.2, 49.9, 5.8, 1.2, and 2.2 mg/100g of sterol, vitamin C, manganese, and vitamins B1 and B5, respectively. These simulated results of nutrient intake resemble the best the estimated intake results of treatments BP1 as 223.0 ± 14.8, 55.93 ± 14.94, 2.34 ± 0.17, 0.98 ± 0.05, and 1.67 ± 0.04 mg/100g and

**Table 6. Estimated nutrient intake in dry-weight basis by self-selecting crickets in seven treatments with different combinations of ingredients.**

| Nutrient | Treatments | | | | | | |
|---|---|---|---|---|---|---|---|
| | NP1 | NP2 | BP1 | BP2 | BP3 | AL1 | AL2 |
| **Macro-Nutrients (mg/g)** | | | | | | | |
| Lipid | 101.9 ± 3.9 | 118.7 ± 9.3 | 54.7 ± 1.1 | 56.3 ± 1.6 | 81.4 ± 2.2 | 96.5 ± 4.2 | 174.8 ± 4.3 |
| Protein | 255.0 ± 8.4 | 311.2 ± 11.1 | 272.7 ± 5.4 | 295.5 ± 4.0 | 283.6 ± 5.1 | 259.1 ± 14.3 | 294.2 ± 4.4 |
| Carbohydrate | 604.7 ± 9.1 | 542.1 ± 18.9 | 594.2 ± 4.5 | 567.6 ± 8.0 | 556.1 ± 4.6 | 494.1 ± 13.3 | 425.6 ± 2.8 |
| Fiber (ND) | 246.3 ± 11.1 | 251.1 ± 18.3 | 280.6 ± 4.8 | 263.7 ± 6.4 | 269.1 ± 3.0 | 186.4 ± 10.8 | 257.1 ± 4.7 |
| **Sterols (mg/100 g)** | | | | | | | |
| Cholesterol | 25.5 ± 1.87 | 0 | 0 | 0 | 0 | 0.22 ± 0.02 | 0.24 ± 0.01 |
| Phytosterol | 0 | 87.81 ± 10.15 | 14.65 ± 3.91 | 21.89 ± 2.61 | 333.8 ± 17.6 | 30.03 ± 2.88 | 526.2 ± 37.7 |
| Ergosterol | 157.8 ± 27.0 | 243.7 ± 12.7 | 208.3 ± 16.0 | 261.4 ± 14.4 | 228.5 ± 21.6 | 0 | 0 |
| Total Sterol | 183.3 ± 28.0 | 331.5 ± 17.4 | 223.0 ± 14.8 | 283.2 ± 13.5 | 562.2 ± 26.2 | 30.2 ± 2.9 | 526.4 ± 37.7 |
| **Vitamins (mg/100 g)** | | | | | | | |
| A[1] | 2.71 ± 0.16 | 0.49 ± 0.02 | 11.69 ± 0.47 | 14.06 ± 0.78 | 10.08 ± 0.28 | 7.84 ± 0.94 | 8.65 ± 0.5 |
| B-Carotene | 3.34 ± 0.43 | 0.03 ± 0.001 | 0.63 ± 0.27 | 0.08 ± 0.01 | 0.42 ± 0.06 | 0 | 0.06 ± 0.003 |
| E | 3.59 ± 0.39 | 5.82 ± 0.55 | 2.4 ± 0.18 | 2.11 ± 0.12 | 2.82 ± 0.05 | 1.34 ± 0.13 | 3.72 ± 0.07 |
| C | 2.26 ± 0.17 | 0.22 ± 0.03 | 55.93 ± 14.94 | 83.57 ± 9.96 | 60.02 ± 7.22 | 0.05 ± 0.02 | 1.88 ± 0.1 |
| K | 0.23 ± 0.03 | 0.002 ± | 0.14 ± 0.02 | 0.15 ± 0.02 | 0.13 ± 0.01 | 0.0002 | 0.005 ± |
| B1 | 0.84 ± 0.08 | 1.37 ± 0.04 | 0.98 ± 0.05 | 1.15 ± 0.04 | 1.51 ± 0.07 | 1.54 ± 0.12 | 1.46 ± 0.06 |
| B2 | 1.58 ± 0.09 | 1.42 ± 0.05 | 1.5 ± 0.08 | 1.67 ± 0.05 | 1.5 ± 0.08 | 0.63 ± 0.1 | 1.11 ± 0.03 |
| B3 | 9.82 ± 0.68 | 14.29 ± 0.59 | 12.73 ± 0.27 | 13.35 ± 0.33 | 17.98 ± 0.49 | 19.36 ± 1.42 | 17.19 ± 0.51 |
| B4 | 91.6 ± 3.9 | 78.0 ± 3.5 | 152.9 ± 3.4 | 166.7 ± 9.5 | 175.3 ± 9.5 | 152.9 ± 9.2 | 217.8 ± 13.6 |
| B5 | 2.42 ± 0.04 | 1.88 ± 0.04 | 1.67 ± 0.04 | 1.56 ± 0.01 | 2.66 ± 0.06 | 0.92 ± 0.04 | 3.32 ± 0.14 |
| B6 | 0.96 ± 0.05 | 1.4 ± 0.04 | 0.81 ± 0.04 | 0.84 ± 0.03 | 1.5 ± 0.04 | 1.81 ± 0.17 | 1.4 ± 0.08 |
| B9 | 0.31 ± 0.03 | 0.19 ± 0.01 | 0.2 ± 0.02 | 0.2 ± 0.01 | 0.2 ± 0.01 | 0.07 ± 0.003 | 0.07 ± 0.001 |
| **Minerals (mg/100 g)** | | | | | | | |
| Ca | 556.4 ± 36.1 | 78.0 ± 3.7 | 154.2 ± 17.5 | 133.1 ± 10.9 | 183.7 ± 15.1 | 1868.5 ± 167.3 | 168.6 ± 8.9 |
| Fe | 13.97 ± 1.2 | 7.32 ± 0.29 | 7.33 ± 0.68 | 5.47 ± 0.14 | 9.78 ± 0.35 | 15.97 ± 0.78 | 16.86 ± 0.24 |
| Mg | 300.3 ± 15.0 | 372.8 ± 20.2 | 282.9 ± 6.6 | 241.8 ± 2.6 | 361.3 ± 8.8 | 622.0 ± 32.8 | 456.1 ± 9.3 |
| P | 581.4 ± 19.8 | 646.5 ± 25.8 | 562.5 ± 17.0 | 517.1 ± 16.9 | 753.9 ± 22.6 | 1562.3 ± 62.3 | 1057.9 ± 16.6 |
| K | 1526.1 ± 17.0 | 1530.0 ± 30.1 | 1437.2 ± 56.7 | 1560.9 ± 43.2 | 1598.6 ± 27.0 | 1482.6 ± 20.8 | 1397.1 ± 5.6 |
| Na | 137.3 ± 9.7 | 53.1 ± 2.2 | 125.6 ± 4.4 | 154.6 ± 4.6 | 127.7 ± 3.4 | 178.5 ± 15.2 | 371.6 ± 10.1 |
| Zn | 4.56 ± 0.13 | 5.13 ± 0.27 | 4.49 ± 0.07 | 4.27 ± 0.06 | 4.8 ± 0.05 | 7.25 ± 0.07 | 6.06 ± 0.07 |
| Cu | 1.21 ± 0.04 | 2.36 ± 0.13 | 1.27 ± 0.04 | 1.31 ± 0.04 | 1.25 ± 0.05 | 0.81 ± 0.03 | 0.52 ± 0.006 |
| Mn | 3.98 ± 0.33 | 5.46 ± 0.5 | 2.34 ± 0.17 | 1.26 ± 0.02 | 3.95 ± 0.15 | 9.53 ± 0.65 | 5.92 ± 0.24 |

Mean ± standard deviation.
[1]I.U./g

treatment BP3 as 562.2 ± 26.2, 60.02 ± 7.22, 3.95 ± 0.15, 1.51 ± 0.07, and 2.66 ± 0.06 mg/100g of sterol, vitamin C, manganese, and vitamins B1 and B5, respectively (Table 6). The mean ending live biomass per experimental unit was 149.0 ± 37.8 and 197.8 ± 70.0 for treatments BP1 and BP3, respectively and these values were the highest observed among the seven treatments. Estimated intake values of vitamin C were substantially lower in treatments NP1, NP2, AL1, and AL2 and higher in treatment BP2. Sterol estimated intake was substantially lower in treatments AL1 and NP1 and manganese estimated intake was substantially higher in treatments NP2, AL1 and AL2 (Table 6).

**Table 7. Results of the stepwise regression analysis of the impact of nutrient intake on final live cricket biomass.**

| Step | Parameter entered | Added Sum of Squares | $R^2$ | $Cp$ | BIC | p |
|---|---|---|---|---|---|---|
| 1 | Vitamin C | 109270.2 | 0.4158 | 55.84 | 711.5 | 2 |
| 2 | Vitamin B5 | 25293.47 | 0.5121 | 38.42 | 703.81 | 3 |
| 3 | Protein | 20163.5 | 0.5888 | 24.95 | 696.7 | 4 |
| 4 | Sterol | 15308.06 | 0.647 | 15.2 | 690.81 | 5 |
| 5 | Vitamin B1 | 6206.5 | 0.6707 | 12.43 | 690.43 | 6 |
| 6 | Manganese | 15418.0 | 0.7293 | 2.6* | 681.67* | 7* |
| 7 | Selenium | 3516.66 | 0.7427 | 1.9 | 682.51 | 8 |
| 8 | Lipid | 467.7 | 0.7445 | 3.54 | 686.25 | 9 |
| 9 | Vitamin B4 | 292.76 | 0.7456 | 5.32 | 690.15 | 10 |
| 10 | Vitamin A | 4710.86.52 | 0.7635 | 3.7 | 689.51 | 11 |
| 11 | Vitamin B9 | 1630.12 | 0.7697 | 4.45 | 691.95 | 12 |
| 12 | Magnesium | 2058.91 | 0.7776 | 4.87 | 693.85 | 13 |
| 13 | Zinc | 379.31 | 0.779 | 6.58 | 697.61 | 14 |
| 14 | Vitamin E | 752.76 | 0.7819 | 8.0 | 700.94 | 15 |
| 15 | Calcium | 308.85 | 0.783 | 9.76 | 704.78 | 16 |
| 16 | Vitamin B2 | 1634.8 | 0.7893 | 10.51 | 707.05 | 17 |

## Phase 2

The estimated macro nutrient ratios of Paton's 13 diet were 0.062 lipid (L), 0.268 protein (P), and 0.67 carbohydrate (C). These ratios were slightly more carbohydrate biased than those in the new diets but fell within the macro nutrient ratios observed in the self-selection experiment among different treatments. Macro nutrient ratios for diets 1 to 4 were 0.06 L: 0.3 P: 0.64 C, 0.075 L: 0.297 P: 0.628 C, 0.091 L: 0.308 P: 0.595 C, and 0.105 L: 0.3 P: 0.595 C, respectively. Macro nutrient ratios in the commercial diet were not determined due to lack of sufficient information on the formulation.

**Large group experiment.** The reference diet Patton's diet 13 was the overall best performer producing significantly more live biomass ($F = 5.46$; df 5, 66; $P = 0.0003$) and dry-weight biomass ($F = 7.38$; df 5, 66; $P < 0.0001$) per experimental unit than all the other diets tested (Fig 4). The efficiency of conversion of ingested food (ECI) was significantly different only between the Patton 13 and the commercial diets ($F = 4.28$; df 5, 66; $P = 0.002$) where the Patton diet was higher. There was not significant difference in ECI among the rest of the diet treatments. Crickets fed on the Patton 13 diet assimilated food significantly better than those feeding diets 2, 3 and 4 ($F = 5.37$, df 5, 66; $P = 0.0003$) (Fig 4). However, there was not significant difference among diet treatments in efficiency of conversion of assimilated food (ECA).

**Table 8. Model from stepwise on live biomass.**

| Parameter | Estimate | Sum of Squares | F ratio | P > F |
|---|---|---|---|---|
| Sterol | 0.828 ± 0.164 | 330809.6 | 25.55 | < 0.0001 |
| Vitamin B1 | -344.06 ± 82.08 | 21183.2 | 17.57 | < 0.0001 |
| Vitamin C | 1.61 ± 0.44 | 16341.3 | 13.55 | 0.0005 |
| Manganese | 35.96 ± 10.05 | 15418.0 | 12.79 | 0.0007 |
| Vitamin B5 | -96.96 ± 28.89 | 13575.9 | 11.26 | 0.0014 |
| Protein | -0.014 ± 0.004 | 11396.0 | 9.45 | 0.0032 |

Model: $R^2 = 0.729$; $F = 26.5$; df 6, 59; $P < 0.0001$.

**Table 9. Optimal model from stepwise with interactions and quadratic effects.**

| Parameter | Estimate | Sum of Squares | F ratio | P > F |
|---|---|---|---|---|
| Vitamin B1 | -753.46 ± 110.5 | 42264.6 | 46.5 | < 0.0001 |
| Sterol | 1.43 ± 0.23 | 35068.6 | 38.58 | < 0.0001 |
| Vitamin C | 1.61 ± 0.36 | 18439.1 | 20.28 | < 0.0001 |
| Vitamin B5 | -98.27 ± 30.88 | 9208.8 | 10.13 | 0.0024 |
| Manganese | 28.86 ± 9.62 | 8179.8 | 9.0 | 0.004 |
| B5 x B5 | 44.96 ± 16.97 | 6378.1 | 7.02 | 0.0104 |
| B5 x Mn | -39.98 ± 7.52 | 25651.6 | 28.22 | < 0.0001 |
| Sterol x B1 | -1.13 ± 0.26 | 16656.6 | 18.32 | < 0.0001 |

Model: $R^2 = 0.803$; $F = 29.01$; df 8, 57; $P < 0.0001$.

No significant differences were observed in the number of live crickets at the end of the experiment among diet treatments; however, the percentage of adult crickets was significantly higher on the Patton diet 13 than in the other diet treatments ($F = 15.97$; df 5, 66; $P < 0.0001$), with the exception of the commercial diet, which ended with similar percentage of adults as the Patton 13 diet (Fig 5). The mean adult weight was significantly higher in the Patton 13 diet treatment than in the other treatments, except for diet 3 treatment, which show no significant differences in adult weight with the Patton 13 diet treatment (Fig 5).

In general, there was no significant difference in performance among diet treatments 1 to 4 and the commercial diet in the large group experiment. The only exception was that the percentage of adults was higher in the commercial diet treatment than in diet treatments 1 and 2 (Fig 5). When the GLM model included percentage of adults as an independent variable, the ending live biomass and dry weigh biomass did not differ significantly among the diet treatments including the Patton 13 diet. This seems to indicate that the higher biomass gain in the Patton 13 treatment was a result of a higher developmental speed and adult growth.

**Small group experiment.** Results of the small group experiment are presented as least square means ± standard error because tray position inside chambers had a significant impact on adult weight and development time. As a result, variables dealing with block distribution had to be included in the general linear model during the analyses.

Diet treatment significantly impacted adult live weight ($F = 33.86$; df 5, 831; $P < 0.0001$), adult dry weight ($F = 57.09$; df 5, 831; $P < 0.0001$), and development time ($F = 168.28$; df 5, 831; $P < 0.0001$) of the house crickets. Sex had significant impact on adult dry weight ($F = 22.4$; df 1, 831; $P < 0.0001$), females having higher dry weight (73.6 ± 0.86 mg) (least square mean ± standard error) than males (68.26 ± 0.73 mg). Sex also impacted development time significantly ($F = 159.18$; df 1, 831; $P < 0.0001$), females developing faster (52.17 ± 0.23 days) than males (56.06 ± 0.2 days). Adult live weight did not differ significantly between females (253.83 ± 2.4 mg) and males (256.67 ± 2.02 mg).

The live weight of adult crickets was significantly higher in the Patton 13 (286.21 ± 3.49 mg) and commercial (274.96 ± 3.48 mg) diet treatments as compared with that of diets 1 to 4. Adult crickets of diet 2 had a significantly lower live weight (228.34 ± 3.98 mg) than the rest of the treatments. There was no significant difference in adult live weight between the Patton 13 and commercial diet treatments and among diet treatments 1, 3, and 4 (245.4 ± 3.79, 251.23 ± 3.91, and 245.37 ± 4.0 mg, respectively). Results were similar for the adult dry weight, except that there were significant differences in adult dry weight between Patton 13 (86.68 ± 1.26 mg) and commercial (77.52 ± 1.25 mg) diet treatments. The longest development time was observed in the diet 2 treatment (61.43 ± 0.39 days), which was significantly higher than that

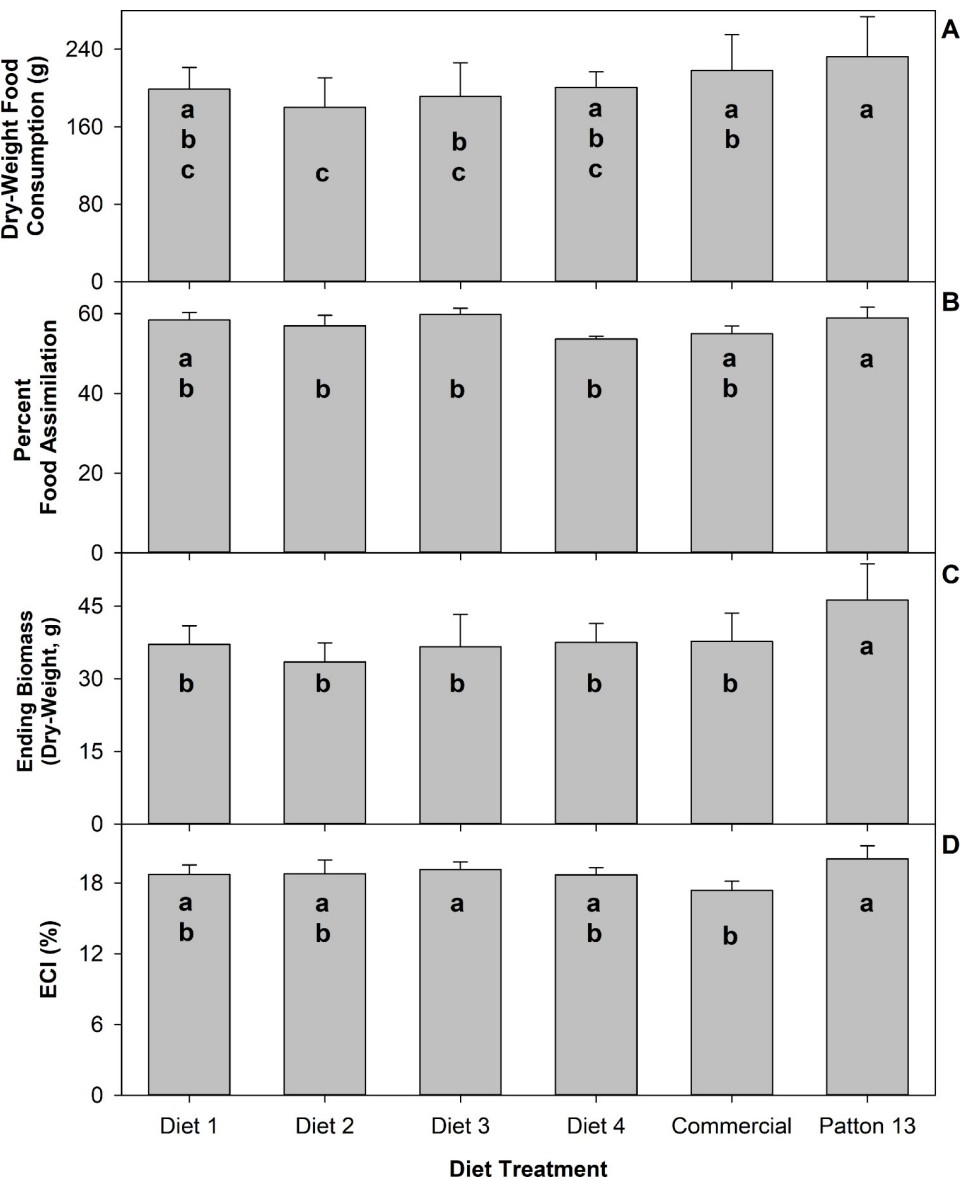

**Fig 4.** Means of dry-weight food consumption in g (A), percent food assimilation (B), ending dry-weight biomass gained in g (C), and percent efficiency of conversion of ingested food (ECI) (D). Brackets represent standard deviation. Columns with the same letter are not significantly different after GLM's Tukey-Kramer HSD test of LS means at α = 0.05.

of the rest of the treatments. The diet 4 treatment showed the second longest development time (57.16 ± 0.39 days), which was significantly higher than that of the rest of the treatments except for diet 2. Development time in the Patton 13 (49.48 ± 0.34 days) and commercial (48.71 ± 0.34 days) diet treatments was significantly shorter than that of the other diet treatments. Development time between the diet 1 (54.0 ± 0.37 days) and diet 3 (53.91 ± 0.38 days) treatments did not differ significantly. Diet treatment also affected cricket survival from first instar to adult significantly ($X^2$ = 40.97; df = 5; $P < 0.0001$) (Fig 6). The Patton 13 (0.815 ± 0.0275) (mean ± standard deviation) and commercial (0.815 ± 0.0275) diet treatments had significantly higher survival than the rest of the diet treatments. The diets 2 and 4 showed

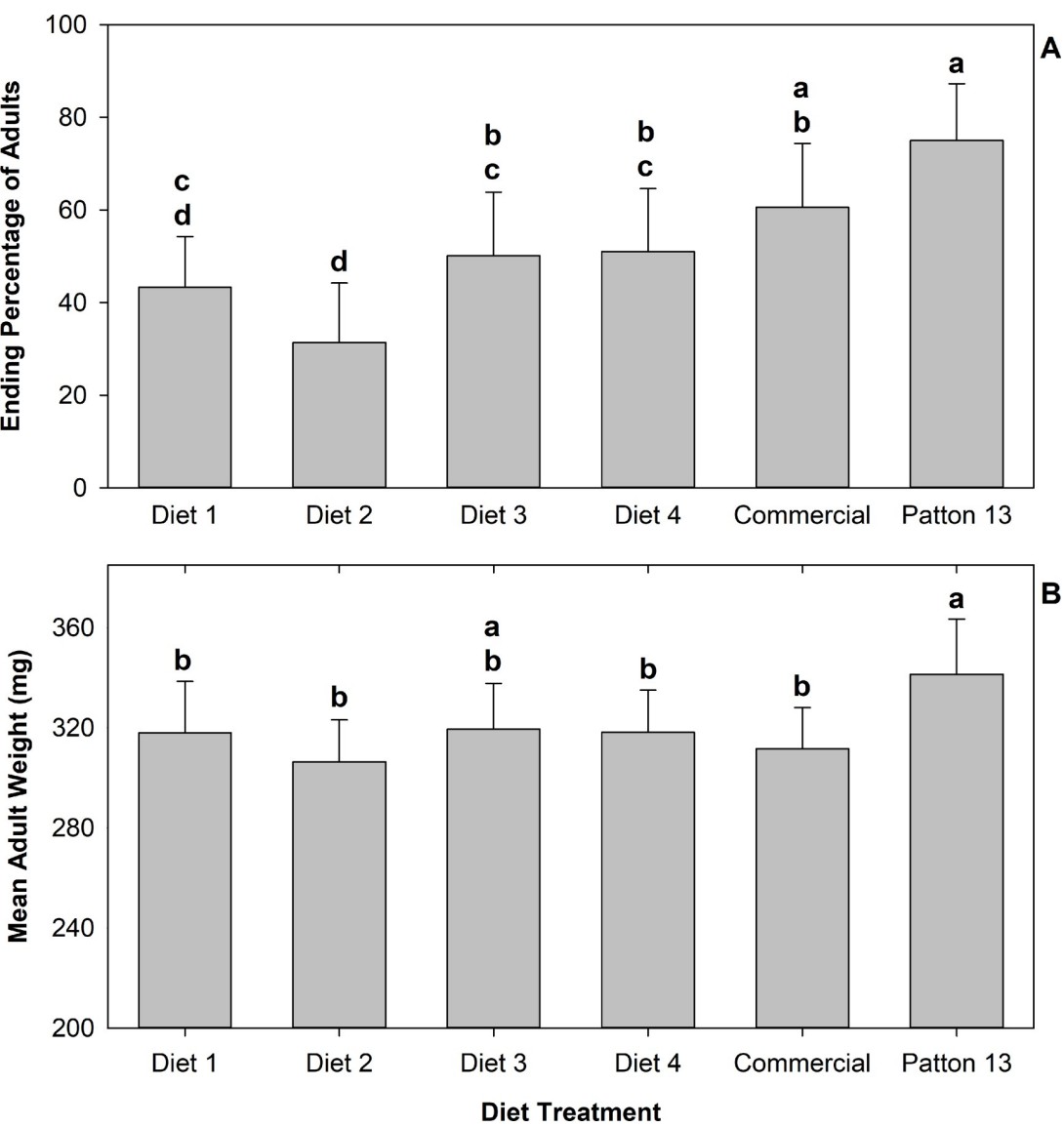

**Fig 5.** Means of ending percentage of adults (A) and adult individual weight in mg (B) of crickets reared in 6 different diets. Brackets represent standard deviation, bars with the same letter are not significantly different after Tukey-Kramer HSD test at α = 0.05.

significantly lower survival (0.625 ± 0.0342 and 0.62 ± 0.0344, respectively) than the rest of the treatments (diets 1 and 3 survival was 0.69 ± 0.0328 and 0.65 ± 0.0338, respectively) (Fig 6). There were some interactions between diet treatment and sex in adult weight and development, generating slightly different analysis results between sexes (Fig 7).

In general, the reference diet Patton 13 and the commercial diet performed better by producing larger adults, which developed faster, and survived better than adults produced by the new diet formulations (1, 2, 3, and 4) in the small group experiment. Among the new diet formulations, diet 2 was the poorest performer producing smaller adults and developing slower than adults produced by diets 1, 3, and 4. There were some discrepancies between the large and small group experiments. For instance, the commercial diet did not differ in cricket biomass production and ending percentage of adults as compared to the new diet formulations in

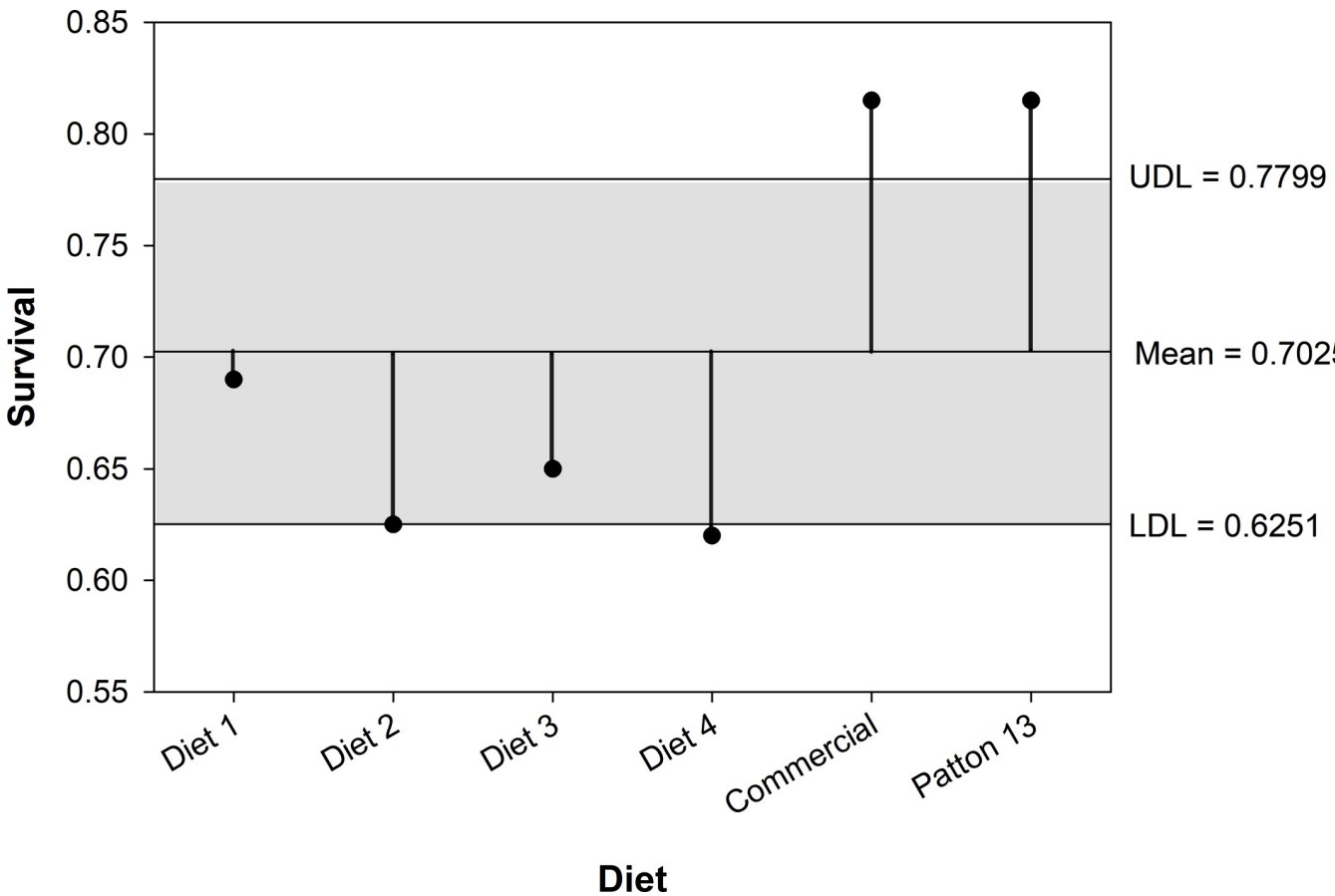

**Fig 6. Analysis of means for proportions of live crickets after developing in 6 different diets.** UDL = upper difference limit, LDL = lower difference limit. Proportions outside the difference limits are significantly higher (UDL) or lower (LDL) than proportions within the limits.

the large group experiment, but in the small group experiment, the commercial diet produced larger adults, which developed faster than those produced by the new diet formulations. In general, it seems that the commercial diet performed better in small groups than in large groups.

**Economic analysis.** Based on current ingredient prices, diet 4 was the cheapest with an estimated price of $0.39 USD per kg and the commercial diet was the most expensive with a retail price of $5.10 USD per kg, but this price includes costs for labor and shipping of ingredients to the milling site. For this reason, the commercial diet could not be compared with the other diets in the economic analysis. The estimated cost per kg of all the new diets was lower than the reference (Patton 13) diet (Table 10). The average price of 1 kg of cricket powder was $93.05 ± 14.01 USD from 9 different companies obtained in April 2019. The estimated revenue per kg of cricket powder and per $m^3$ or rearing space using the 6 diets based on diet pricing, cricket powder pricing, and food conversion results from the large group experiments, is presented in Table 10. The estimate revenue from the five diets included in the analysis do not consider labor and ingredient shipment costs, which can variate widely depending of technology used and country of origin.

Patton's 13 diet cost was much higher and the estimated revenue per kg of cricket power was lower than those of obtained for the four new diets. However, when the revenue is calculated in function of rearing space, Patton's 13 diet economic performance was slightly better

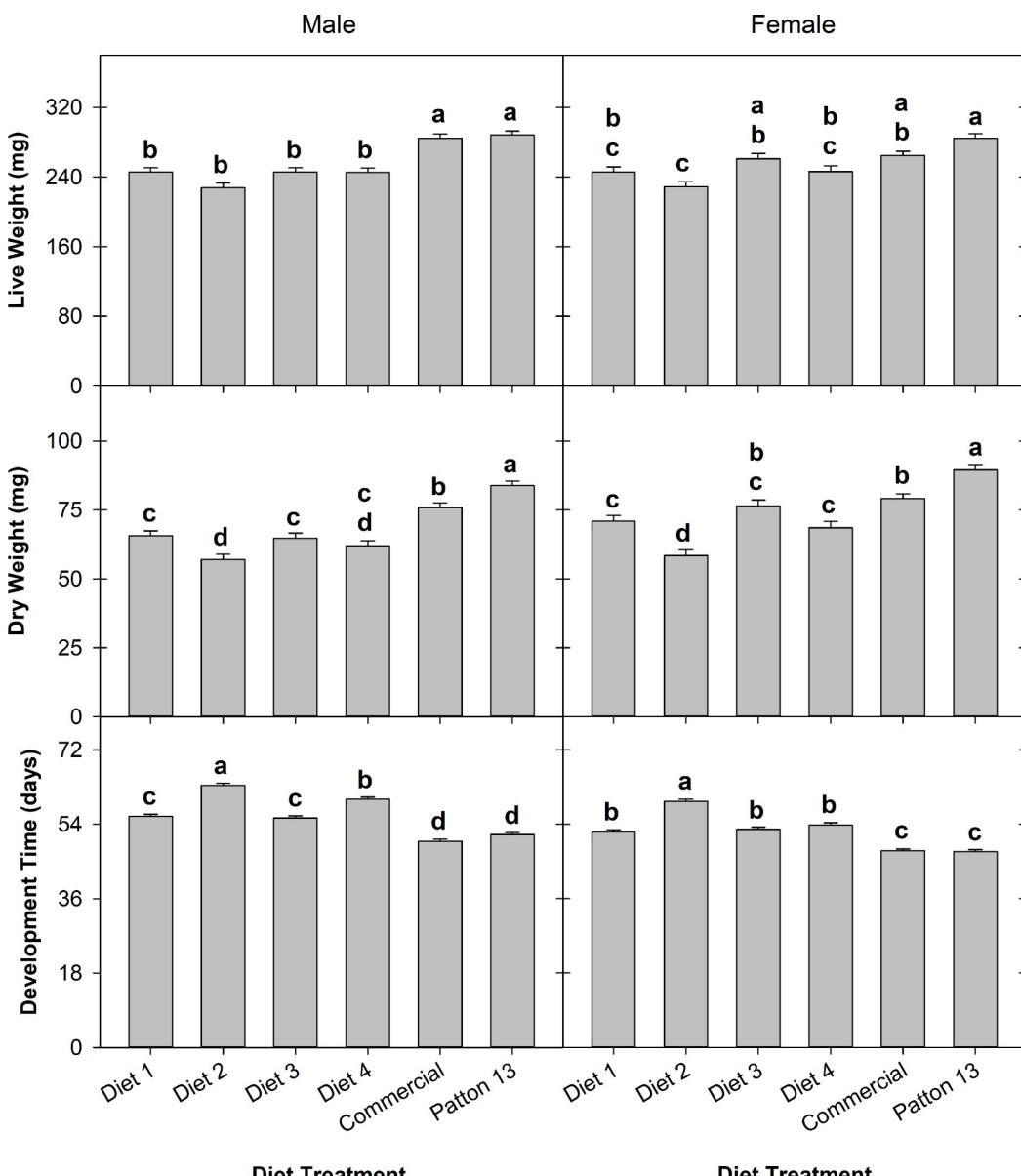

**Fig 7. Least square means of live weight, dry weight, and development time of male and female crickets feeding on 6 different diets.** Brackets represent standard error, bars with the same letter are not significantly different after Tukey-Kramer HSD test for LS means at $\alpha$ = 0.05.

than diets 1, 2, and 3. Only diet 4 produced more revenue per $m^3$ of rearing space than Patton's 13 diet (Table 10). These results are most likely due to the current high price of cricket powder in combination with the high productivity of crickets reared using Patton's 13 diet.

## Discussion

Results of the self-selection experiments demonstrated that house crickets have preferences for some ingredients over others and that these preferences change depending on the choices available (Tables 3 and 4). However, consumption of different ingredients at different ratios resulted in a high degree of convergence in the macro-nutrient intake ratios. Deviations from

**Table 10. Economic analysis of cricket production using five different diets.** Costs and revenues in USD. Cricket dry-weight biomass productivity was based on food utilization results from the large group experiment.

| | Diet 1 | Diet 2 | Diet 3 | Diet 4 | Patton 13 |
|---|---|---|---|---|---|
| Price per kg of diet[1] | $0.87 | $0.64 | $0.75 | $0.39 | $3.54 |
| Diet cost per kg of cricket powder | $4.71 | $3.56 | $3.99 | $2.07 | $18.20 |
| Diet cost per $m^3$ of rearing space per cycle | $8.90 | $6.14 | $7.31 | $4.15 | $41.11 |
| Revenue per kg of cricket powder[2] | $88.34 | $89.49 | $89.06 | $90.98 | $74.85 |
| Dry biomass in kg per $m^3$ of rearing space | 1.89 | 1.72 | 1.83 | 2.01 | 2.26 |
| Revenue per $m^3$ of rearing space per cycle[2] | $167.15 | $154.35 | $163.35 | $182.65 | $169.04 |
| Revenue per $m^3$ of rearing space per cycle/year[2] | $835.76 | $771.74 | $816.75 | $913.27 | $845.22 |

[1]Pricing of diet formulations based on ingredient pricing obtained on line in April 2019 (AgEBB 2019, Alibaba.com 2019). Price of the commercial diet was based on retail list price.

[2]Average cricket powder price of $93.05 ± 14.01 USD from 9 different companies.

these ratios, as it occurred in treatments NP2 and AL2, resulted in significant reduction of ending live biomass gain. Simple regression models on ending cricket biomass showed that carbohydrate content had a significant positive effect, while lipid content had a significant negative effect and protein had no effect. However, in the multiple regression response surface analysis the content of macronutrients had no significant impact on ending biomass. The reason for the low degree of impact of macro nutrient intake on biomass shown in the regression analyses is most likely due to the small range of variation in macro nutrient intake resulting from the self-selection experiment. Self-selecting by the crickets limited the macro nutrient intake ratios within a restricted range of values avoiding extreme biases of any of the three macro nutrients, which could have impacted cricket growth and biomass production.

Other nutrients including vitamin C, sterol, and manganese had significant positive impact on ending live biomass. High consumption of some food ingredients like rice bran, wheat bran, cabbage, brewer's yeast and spirulina may be associated to high content of one or more of these three nutrients. For instance, rice bran has high content of phytosterols and manganese, brewer's yeast has high content of ergosterol, cabbage and spirulina contain vitamin C and wheat bran has high content of manganese. Although evidence of nutrient self-selection in most insect species focuses on macro-nutrient ratios, there is evidence that insects can select for optimal vitamin ratios as well [50]. Other study has provided evidence that some insects can also select optimal ratios of salt [51]. The mechanisms for simultaneous self-selection for multiple nutrients are expected to be complex due to potential interactions among nutrients [52]. Scarcity of some essential nutrients may drive consumption of a food ingredient relatively rich in one of these nutrients, disbalancing other nutrients less limiting in the process as a tradeoff [52]. This phenomenon may be responsible for the macro nutrient disbalance observed in treatments NP2 and AL2.

Vitamin C and sterol are known to be essential in all insects, with some exceptions, and both play important roles in the molting process [18, 53, 54, 55, 56]. Vitamin C is present in cabbage, whole dry milk, spirulina, and sunflower kernels [22]. Sterol is present in brewer's yeast as ergosterol [26], in cabbage, sunflower kernels, and rice bran as phytosterols, and in whole dry milk as cholesterol [22]. High consumption of these ingredients may be explained by their content of vitamin C and sterol. For instance, high ratios of lipid intake in treatments NP2 and AL2 could be explained by high consumption of sunflower kernels and rice bran, respectively in absence of other sources of sterol and vitamin C. Similarly, high consumption of whole milk powder may have driven high intake ratios of carbohydrate in treatment NP1. High consumption of brewer's yeast in treatments NP1, NP2, BP1, BP2, and BP3 may have

contributed to an excessive consumption of B vitamins explaining the negative impact of vitamins B1 and B5 on ending live biomass. Manganese is present in many of the ingredients tested in this study, but wheat and rice brans contain the highest amount of this element [22], explaining the high consumption observed of these two ingredients, especially the defatted rice bran consumption of 50.8% in treatment AL1.

Diets 1 and 3 were formulated based on self-selection of ingredients from treatments BP1 and BP3, respectively. Although these diets performed well and crickets fed on these diets produced dry-weight biomass similar to that produced by crickets fed the commercial diet, the self-selected formulations did not match the productivity of Patton's diet 13. In general, Patton's diet 13 performed better than all diets tested.

The estimated content of vitamin C in diets 1 to 4 was 39.6, 0.0, 44.0, and 35.2 mg/100g, respectively. Patton's 13 diet had an estimated content of vitamin C of 1.6 mg/100g. It seems that vitamin C content could not explain performance differences observed between the new diets and Patton's 13 diet. Also, it seems that vitamin C is important, but crickets were able to develop and growth in its absence, since diet 2 had no vitamin C. Crickets feeding on diet 2 completed development in a significantly longer time than all other diets, but dry-weight biomass gain was not significantly lower than the other three new diets in the large group experiment.

The estimated sterol (sum of cholesterol + phytosterol + ergosterol) content in diets 1 to 4 was 202.4, 208.2, 515.2, and 468.5 mg/100g, respectively, and the estimated sterol content of Patton's 13 diet was 89.7 mg/100g. Since the content of sterol in Patton's 13 diet is estimated to be a fraction of that in diets 1 to 4, sterol content does not explain the high performance of Patton's 13 diet. Differences in estimated content of manganese could not explain Patton's performance either. Diet 4 had a higher estimated Mn content of 6.9 mg/100g than that of Patton's diet 13 of 4.0 mg/100g and diet 3 had an estimated Mn content of 3.6 mg/100g. Although estimated Mn content of diet 1 and 2 was lower, there were not significant differences in dry-weight biomass gain and development time among diet 1, 3 and 4. Estimated levels of other important vitamins including vitamin A, E, $B_1$, $B_2$, $B_4$, $B_6$, $B_5$, and $B_9$ were all similar or higher in the new diet formulations as compared to those of the Patton's diet 13.

The main difference between the new diet formulations and Patton's diet 13 is the presence of ingredients of animal origin (beef liver and dry milk) in the latter. Patton (1967) [11] observed that the better performing diet formulations contained either liver powder or fish meal. The author concluded that a growth factor was present in liver powder, meat scraps and menhaden fish meal responsible for the better performance of those diets based on a previous report by Neville et al. (1961) [57]. Our results showed that crickets fed on Patton's diet 13 and the commercial diet had significantly faster development than crickets fed on the new diets. The commercial formulation is reported to contain animal scraps by the manufacturer, supporting Patton (1967) [11] conclusion. Ingredients of animal origin usually contain vitamin $B_{12}$, which is absent in ingredients of vegetable origin. In our study the only source of vitamin $B_{12}$ was brewer's yeast, but the content of this vitamin in brewer's yeast is only 10 ppb (parts per billion = μg/kg) while dry beef liver contains 1,980 ppb. In addition, dry milk contains 40 ppb of vitamin $B_{12}$. Patton's 13 diet is estimated to contain 105 ppb of vitamin $B_{12}$, while diets 1 to 4 are estimated to contain less than 2 ppb of this vitamin. Vitamin $B_{12}$ was not included in the multiple regression analyses because the ingredients tested lack it or contain only trace amounts of it.

It is obvious that vitamin $B_{12}$ is not required in the diet of the house cricket because they are able to complete development and reproduce in the absence of vitamin $B_{12}$ dietetic sources (such as in treatments AL1 and AL2). Nevertheless, adult crickets are known to contain between 53.7 and 193 ppb of vitamin $B_{12}$ [43, 58]. This vitamin could be synthesized by the symbiotic microflora of the house crickets. Ulrich et al. (1981) [59] reported 25 species of bacteria from the genera *Citrobacter*, *Klebsiella*, *Yersinia*, *Bacteroides*, and *Fusobacterium* living in

the midgut and hindgut of *A. domesticus*. Germ-free *A. domesticus* are reported to be less efficient converting food into biomass [60]. Although no information has been published on the role of the house cricket microflora on the synthesis of vitamins, it is known that *Citrobacter freundii* and *Klebsiella pneumoniae* can synthetize vitamins $B_2$, $B_6$, and $B_{12}$ [61]. However, it is possible that additional sources of vitamin $B_{12}$ provide benefits to cricket nutrition that translate in more efficient food utilization and faster development and growth.

Another important difference between crickets fed with Patton's diet 13 and those fed with the new diets was food consumption. Crickets fed with Patton's diet 13 consumed significantly more food than all the other treatments in the large group experiment and significantly more than crickets feeding on diets 1 to 4 in the small group experiment. Food utilization efficiency expressed as ECI and ECA were not significantly different among treatments, however crickets assimilated Patton's 13 diet significantly better than diets 2, 3, and 4. The differences in biomass production between crickets fed on Patton's 13 diet and diets 1 to 4 may be the result of higher food consumption and better food assimilation rather than better diet quality. It is possible that crickets grow faster and larger in Patton's diet 13 because they eat more of it faster. In this case, feeding stimulants [17, 18] may be responsible for the beneficial effects of ingredients of animal origin. Feeding stimulants can be substances with no nutritional value, but that serve as identifiers of suitable food [18]. The advantages in assimilation can be explained by the higher content of more digestible forms of carbohydrate, such as sugars and starch in Patton's diet 13, and higher content of less digestible carbohydrates like fiber in diets 1 to 4.

Adding ingredients of animal origin may enhance the performance of cricket diets but the addition of such ingredients will impact the sustainability value of cricket-based food or feed. It will also increase the environmental impact and carbon footprint of cricket production detracting to some extent from the value of insect-based feed and food as a sustainable and environmentally friendly source of animal protein. Another important value of insects as a source of animal protein for animal feed is the reduced risk of pathogen transmission as diseases affecting insects are not expected to be pathogenic to vertebrates. Inclusion of ingredients of vertebrate origin in insect diets may compromise this advantage as well.

Despite the superior performance of Patton's 13 diet, the high cost of liver powder, reduced its economic potential. The economic analysis showed that the feed cost per kg of cricket powder was $18.00 USD for the Patton's 13 diet compared with the feeding cost for diets 1 to 4 of $4.71, $356, $3.99, and $2.07 USD, respectively. Feeding costs per $m^3$ of rearing space were even higher for Patton's 13 diet ($41.11) compared with that for diets 1 to 4 ($8.90, $6.14, $7.31, and $4.15, respectively). The most economical formulation was diet 4. However, revenues per $m^3$ of rearing space were very competitive for Patton's 13 diet with an estimated $169.04 compared with $167.15, $154.35, $163.35, and 182.65 for diets 1 to 4, respectively. The current high prices of cricket powder and the high productivity of Patton's 13 diet (producing more dry-weight biomass per cycle) makes it economically competitive. Reducing the price of cricket powder makes Patton's 13 diet less competitive as revenue by crickets fed in this diet is more affected by cricket powder pricing than that from crickets produced with the new diets. Nevertheless, diet performance seems to have a high impact on its economic viability. Improving performance of by-product-based diet formulations should be the next research goal to increase cricket production revenues.

Overall, diet 4 was the most economically viable diet due to the low price of its formulation. Despite that diet 4 production performance was inferior to Patton's 13 diet, the economic performance of diet 4 was better. This economic analysis did not account for costs of diet mixing and shipment costs of ingredients and adding these costs could change the results. Also, rearing labor costs were not included, and these costs could significantly reduce the levels of revenue calculated in this study. Because of its low cost, however, diet 4 provides the highest level

of flexibility making this formulation highly promising. Diet 4 is considered the best of the new diets based on its performance, cost, and percentage of by-product composition.

The use of self-selection of crude ingredients (no chemically defined) has not been used before as a tool to develop insect diets. Other examples of the use of self-selection in insect diet development had relied on the use of different diets of known chemical composition [19, 20, 21]. There is a high level of uncertainty when the nutrient intake by self-selecting insects is based on published analyses of nutritional composition of crude ingredients. However, data obtained on nutrient intake can still be analyzed and results can be useful. Specially on nutrients that are present in consistent quantities like macronutrients. The use of multiple regression analysis on self-selected nutrient intake ratios has never been used before in conjunction with self-selection methods.

Other methods of multidimensional analysis have been used to refine artificial diets in other insect species with great success [62, 63, 64], but these studies are based on existing diet formulations and are used to optimize ratios of three ingredients at a time on their impact on multiple insect biological parameters and costs using Raubenheimer's right-angled mixture triangle multidimensional analysis [65]. Estimating nutrient intake from self-selected consumption of multiple ingredients using a matrix operation and then analyzing their impact on biomass production, as it was done in this study, is a new concept. The advantages of self-selection methods are that a large number of ingredients can be tested, no pre-existing diet formulations are needed, and resulting ingredient consumption ratios tend to converge within relatively narrow margins from optimal ratios. The main disadvantage of the self-selection method is that this method is not applicable to all species and is better suitable for species that have some degree of omnivory and their food consumption can be measured. Other disadvantages include the tendency of some species to overconsume some nutrients and the presence of feeding stimulants with no nutritional value can drive overconsumption of some ingredients. However, multiple regression analysis can provide information on the impact that overconsumption of some ingredients or nutrients can have on the biological parameters of the species under study.

Results obtained in this study will require further refining but have provided a good starting point that could not have been achieved with other methods within the time span of this study. Although most of the by-products tested in this study are already being used as animal feed by livestock and aquaculture industries, these products constitute new feed sources for insect rearing no recognized previously. The price of these by-products remains low because they are being produced in excess of their current demand, but, this may change in the future as the value of these agricultural by-products as animal feed is more widely recognized. Among the ingredients that showed the greatest self-selected consumption were rice bran (whole and defatted), corn DDGS, buckwheat, and dry cabbage. These five ingredients have not been part of published insect diets and hold great promise for insect diet development in the future.

In summary this study provides: 1) demonstration of the value of dietary self-selection studies in developing oligidic insect diets, 2) probe of the value of self-selection methods in combination with regression analysis of nutrient intake on the study of insect nutrition, 3) the first such study involving farmed edible crickets and agricultural by-products, 4) an economic analysis of feed formulations and economic impact and 5) four new cricket diet formulations, which provide adequate cricket nutrition and contain between 62 and 92% agricultural by-products as ingredients.

## Supporting information

**S1 Fig. Experimental unit setting inside the plastic box.** In this example from treatment BP1, the water dispenser (right up corner) and the radially-distributed food choices (left centre) are

sitting on top of the egg cartons (rearing substrate), Color-coded food choices from the top in clockwise direction include: corn DDGS (grey), buckwheat seed (yellow), brewer's yeast (red), wheat bran (pink), cabbage dry (light green), peanut hulls (brown), and alfalfa pellets (green). (TIF)

**S1 Text. Production of by-products in the United States.**
(DOCX)

**S1 Data.**
(CSV)

**S2 Data.**
(CSV)

**S3 Data.**
(CSV)

**S4 Data.**
(CSV)

## Acknowledgments

We thank Millbrook Cricket Farms, Richland, Mississippi, for donating the crickets to start our stock colony. We also thank Ergon Biofuels, Big River Resources Galva LLC, Riceland Foods Inc., Express Grain Terminals LLC, and ADM Processing Co. For donating samples of by-products for our tests.

## USDA disclaimer statement

Mentions of company and product names in this paper do not constitute a recommendation or endorsement by USDA-ARS.

## Author Contributions

**Conceptualization:** Juan A. Morales-Ramos, Aaron T. Dossey.

**Data curation:** Juan A. Morales-Ramos.

**Formal analysis:** Juan A. Morales-Ramos.

**Funding acquisition:** Aaron T. Dossey.

**Investigation:** Juan A. Morales-Ramos, M. Guadalupe Rojas.

**Methodology:** Juan A. Morales-Ramos.

**Resources:** Aaron T. Dossey, Mark Berhow.

**Supervision:** Juan A. Morales-Ramos.

**Validation:** Juan A. Morales-Ramos.

**Visualization:** Juan A. Morales-Ramos, M. Guadalupe Rojas, Mark Berhow.

**Writing – original draft:** Juan A. Morales-Ramos.

**Writing – review & editing:** Aaron T. Dossey.

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
