## [Decision Letter · Decision Letter 0]

20 Sep 2019

PONE-D-19-22442

Self-Selection of Food Ingredients and Agricultural By-Products by the House Cricket, Acheta domesticus (Orthoptera: Gryllidae): A Holistic Approach to Develop Optimized Diets

PLOS ONE

Dear Dr. Morales-Ramos,

Thank you for submitting your manuscript to PLOS ONE. After careful consideration, we feel that it has merit but does not fully meet PLOS ONE’s publication criteria as it currently stands. Therefore, we invite you to submit a revised version of the manuscript that addresses the points raised during the review process.

We would appreciate receiving your revised manuscript by Nov 03 2019 11:59PM. To enhance the reproducibility of your results, we recommend that if applicable you deposit your laboratory protocols in protocols.io, where a protocol can be assigned its own identifier (DOI) such that it can be cited independently in the future. For instructions see: http://journals.plos.org/plosone/s/submission-guidelines#loc-laboratory-protocols

We look forward to receiving your revised manuscript.

Kind regards,

Nicoletta Righini, PhD

Academic Editor

PLOS ONE

Journal Requirements:

2. Thank you for stating the following in the Competing Interests section: "I have read the journal's policy and the authors of this manuscript have the following competing interests: [The author Aaron T. Dossey as the owner of All Things Bugs LLC is planning to commercialize cricket feeds formulated based on this study, however formulations for commercialization are different than the ones reported in this manuscript] "

**Additional Editor Comments (if provided):**

Based upon the comments of two reviewers and my own reading, I am sorry we cannot accept your manuscript in the current form. The ms is generally well written and the data are sound, but there are some issues that should be addressed before it can be published. Thus please address all the queries raised by the reviewers, using their recommendations as a guide to improve your manuscript.

In particular, I agree with Reviewer 1 in that the Introduction must be considerably shortened and focused. For example, the description of several types of by-products on page 6 should be condensed or put on a Table or Supplementary material. Also, please be more explicit with your objectives, clearly matching questions with experiments/analyses carried out to answer them.

Some additional issues:

- I suggest to end the Abstract with a sentence highlighting the importance of your main findings (it now ends with a result).

- It is not clear to me what ‘mixed effects general linear models’ are. Why mixed effects?

- Lines 335-344: the list of ingredients is quite long, better adding it in Supplementary material

Reviewers' comments:

Reviewer's Responses to Questions

**Comments to the Author**

1. Is the manuscript technically sound, and do the data support the conclusions?

Reviewer #1: Yes

Reviewer #2: Yes

2. Has the statistical analysis been performed appropriately and rigorously? 

Reviewer #1: I Don't Know

Reviewer #2: Yes

3. Have the authors made all data underlying the findings in their manuscript fully available?

Reviewer #1: No

Reviewer #2: Yes

4. Is the manuscript presented in an intelligible fashion and written in standard English?

Reviewer #1: Yes

Reviewer #2: Yes

5. Review Comments to the Author

**Reviewer #1:** The manuscript by Morales-Ramos and colleagues, present a nice series of experiments testing food-ingredients and by-products to find an optimal diet formulation for mass rearing of Acheta domesticus as a source of protein for human consumption. The self-selection approach is interesting and seem useful in the task of finding optimal diet formulations. Overall, the manuscript is well written, and the study could be of interest to artificial diet researchers and managers of mass rearing facilities of insects used as food ingredient. The manuscript could benefit from making some precisions on how experiments were conducted and analyzed (experimental data was not available for reviewers); this and other suggestions for improvement are below.

General comments:

1. The introduction includes many interesting information showing the authors expertise in their science. However, this big amount of information presented in the introduction might be one of its major weaknesses. In its current form, the introduction is large (seven paragraphs) and dense to read (I must read it twice to better understand it). I think the introduction could be greatly benefited from making it easier to follow and to the point. Synthesizing the information currently presented in four paragraphs addressing the following questions could help: (i) What is the problem? (ii) What are the solutions for that specific problem? (iii) What are the limitations of existing solutions? (iv) How do you expect to contribute? Consider that short introductions are often more attractive and readable.

2. Speaking of the introduction, I found a severe lack of citations to support author’s statements (e.g., lines 45, 52, 67, 92, 98, 103, 105, 116). All the sources of information included in the manuscript must be recognized and cited.

3. The first of five objectives listed at the end of the introduction seem odd (L 143-144). Which experiment, exactly, tested "the ability of A. domesticus to select among food ingredients and by-products"?

4. What was the functional idea of the study-system authors had before the beginning of their study and what would be the expected results if that idea was true? Include this as hypothesis and predictions. Having these in the manuscript would improve comprehension of the rationale behind the study.

5. The information about food ingredients suppliers (lines 193-213) can be moved to Table 1, indicating with a superscript the supplier of each food ingredient.

6. The materials and methods section should include the rationale for testing the food ingredients considered in the study, and describe how the experimental diets were prepared/mixed.

7. I found the statistical analyses a bit confusing. I am struggling to understand which were the explanatory and response variables considered in the experiments and how they were measured, and why mixed models were appropriate for the analysis of such variables. It would be extremely helpful for readers if authors explicitly declare the explanatory and response variables considered in their study indicating their measurement scales, and justify why mixed models, regression, etc. were the appropriate analyses for that specific variables and for reaching the objectives of the study.

8. Related to the above, I think readers would be greatly benefited if authors include a graphical representation of the experimental designs in phase 1 and phase 2. Having such a figure would allow readers to better understand how the experimental work was conducted and analyzed.

9. There are inconsistencies in the use of terminology for referring the same concept, e.g., "total consumed food" (L 275), "total food consumed" (L 276) and "total food consumption" (L 286); or "phase 1 (L 151) and "phase one" (L 153). For sake of clarity, be consistent with the use of terminology throughout the manuscript.

10. In the discussion, why the self-selection approach with crude ingredients have not been used before for developing insect diets (L 895-896)? What are the advantages/disadvantages of the self-selection approach against other approaches for the development and optimization of insect diets (e.g., mixture experiments and response surface methods - see Lapointe et al. 2008, Pascacio-Villafán et al. 2017, Huynh et al. 2019)?

Huynh, M. P., Hibbard, B. E., Lapointe, S. L., Niedz, R. P., French, B. W., Pereira, A. E., ... & Coudron, T. A. (2019). Multidimensional approach to formulating a specialized diet for northern corn rootworm larvae. Scientific reports, 9(1), 3709.

Lapointe, S. L., Evens, T. J., & Niedz, R. P. (2008). Insect diets as mixtures: optimization for a polyphagous weevil. Journal of insect physiology, 54(7), 1157-1167.

Pascacio-Villafán, C., Birke, A., Williams, T., & Aluja, M. (2017). Modeling the cost-effectiveness of insect rearing on artificial diets: A test with a tephritid fly used in the sterile insect technique. PloS one, 12(3), e0173205.

11. What diet formulation would authors propose as the best in terms of the variables evaluated, and what ingredients do authors propose to investigate further? Please clarify this where appropriate in the abstract and discussion.

12. Figure legends are rather vague. For instance, legend for Figure 7 (revise “7A” in line 684) should indicate how to interpret the points below and above the mean, and the points in or out the shaded area.

Specific comments:

13. L 69-70, provide examples of expensive commercial feeds with their cost.

14. L 99-100, indicate how much canola oil is produced and cite reference.

15. L 115, define oligidic.

16. L 163, containers had a lid? Describe.

17. L 164, provide dimensions of the cardboard egg cartons.

18. L 171, insert "crickets" before parenthesis.

19. L 178, for how long oviposition devices were offered to crickets?

20. L 190, provide dimensions of the cardboard egg cartons.

21. L 257-260, what is "log"? And revise writing.

22. L 273, "... (mg) of ingredient...".

23. L 275, describe how FC was calculated the first time it is mentioned.

24. L 287, GLM is more often used for generalized linear models.

25. L 362, Table 2 include five diets, not six.

26. L 452-454, the results of "survival among diet treatments" are found in Figure 7, but where are the results of "cricket survival from first instar to adult"?

27. L 453-454, I am unfamiliar to "ANOM", is this correct?

28. L 684, 686, 687, include "mean proportion of" as needed. Include error (SE, IC 95%, other) of the mean proportions.

29. L 762, "three" not "tree".

30. L 743-745, include Fig or Table showing that specific result.

31. L 747, "experiments" or "treatments"? Authors need to be consistent with terminology throughout the manuscript.

32. L 888, the manuscript does not include Figure 9.

33. L 895-896, define "crude ingredients".

**Reviewer #2:** The article refers to a comprehensive research on the development of sustainable and convenient diets for the house cricket based on agricultural by-products. The manuscript relies on a well-designed series of experiments, properly arranged and replicated, based on the self-selection of food ingredient, a distinctive procedure which has never been used with Acheta domesticus. In my opinion this is the most interesting aspect for readers, although the findings of this research cover many more aspects, involving diet formulations, economic costs and incomes, impact of macro and micro nutrients. In brief, results are very significant for the discipline.

In detail, objectives are clearly enounced as well as the rationale of the research; the methodology, although a bit repetitive, is straightforward and experiments are carefully described. I am not familiar with the specific statistical analyses applied here nor with economic analysis but they sound rigorous. The huge amount of data is presented clearly and supports robust results. The only flaw that can be noticed is the nutritional composition of ingredients which was not obtained through chemical analyses but from literature (as commented at the end of the discussion chapter), anyway results evidence the effectiveness of the applied methodology.

Here below minor misprints are detailed and the respective line number.

Line number Suggestions and comments

253 Amend “depending on” instead of “depending of”

273 Edit “of” instead of “if”

420 Edit “HEPA filter” instead of “HEPHA filter”

550 Insert “within columns” between “Means with the same letter” and “are not significantly different….”

641 In other figure captions and table titles you specified the unit of the weight, to be consistent you should specify here “in mg”. Otherwise find a different solution (although mg appears in the Y-axis title).

683-684 I think there is a misprint: (Fig 7A) does not exist, amend it in “(Fig 7)”

736 I think “for” should be deleted.

816 Edit “latter” instead of “later”

837 Edit “from” instead of “form”

838 Edit “Fusobacterium” instead of “Fusobacterioum”

888 Figure 9 is missing!

897 Amend “relied” instead of “relayed”

6. PLOS authors have the option to publish the peer review history of their article (what does this mean?). If published, this will include your full peer review and any attached files.

Reviewer #1: No

Reviewer #2: No

---

## [Author Response · Author response to Decision Letter 0]

15 Oct 2019

Answer to Reviewers:

Editor:

Based upon the comments of two reviewers and my own reading, I am sorry we cannot accept your manuscript in the current form. The ms is generally well written and the data are sound, but there are some issues that should be addressed before it can be published. Thus please address all the queries raised by the reviewers, using their recommendations as a guide to improve your manuscript.

In particular, I agree with Reviewer 1 in that the Introduction must be considerably shortened and focused. For example, the description of several types of by-products on page 6 should be condensed or put on a Table or Supplementary material. Also, please be more explicit with your objectives, clearly matching questions with experiments/analyses carried out to answer them.

The number of paragraphs has been reduced as suggested. The paragraphs describing the by-products has been moved to supplemental materials S1 Text. The objectives have been enumerated to facilitate following them and also have been made more explicit.

Additional Issues:

- I suggest to end the Abstract with a sentence highlighting the importance of your main findings (it now ends with a result).

Three sentences have been added at the end of the abstract to highlight the importance of the main findings

- It is not clear to me what ‘mixed effects general linear models’ are. Why mixed effects?

This has been clarifies in the text and a new citation was added.

- Lines 335-344: the list of ingredients is quite long, better adding it in Supplementary material

In here we followed the recommendation of Reviewer 1 and added this information on the footing of Table 1.

Reviewer 1: The manuscript by Morales-Ramos and colleagues, present a nice series of experiments testing food-ingredients and by-products to find an optimal diet formulation for mass rearing of Acheta domesticus as a source of protein for human consumption. The self selection approach is interesting and seem useful in the task of finding optimal diet formulations. Overall, the manuscript is well written, and the study could be of interest to artificial diet researchers and managers of mass rearing facilities of insects used as food ingredient. The manuscript could benefit from making some precisions on how experiments were conducted and analyzed (experimental data was not available for reviewers); this and other suggestions for improvement are below.

General Comments:

1. The introduction includes many interesting information showing the authors expertise in their science. However, this big amount of information presented in the introduction might be one of its major weaknesses. In its current form, the introduction is large (seven paragraphs) and dense to read (I must read it twice to better understand it). I think the introduction could be greatly benefited from making it easier to follow and to the point. Synthesizing the information currently presented in four paragraphs addressing the following questions could help: (i) What is the problem? (ii) What are the solutions for that specific problem? (iii) What are the limitations of existing solutions? (iv) How do you expect to contribute? Consider that short introductions are often more attractive and readable.

The issues associated with the size and organization of the introduction have been addressed.

2. Speaking of the introduction, I found a severe lack of citations to support author’s statements (e.g., lines 45, 52, 67, 92, 98, 103, 105, 116). All the sources of information included in the manuscript must be recognized and cited.

Citation for line 42 was actually the same as for line 47. Some journals do not favor the over use of citations if they are the same in adjacent sentences; however, we have added this citation again in the previous sentence.

The sentence in lines 51-52 has been deleted. This was the only sentence that was actually unsupported.

Citation for line 67 is also [1] and it was added. Comment on sales as novelty food has been deleted.

Citation for line 92 is the same as the citation for next sentence. However, this paragraph has been deleted and added to S1 Text.

The same case applies for citations in lines 98, 103, and 105. In all these cases the citation is present, but not mentioned in every sentence when adjacent to each other, following formats of other journals. Since PlosONE did not specify that repeated citations must be added to each sentence, this was not done. Now it has been corrected

Citation for line 116 is the same as that for line 117. Additional citation was added.

3. The first of five objectives listed at the end of the introduction seem odd (L 143-144). Which experiment, exactly, tested "the ability of A. domesticus to select among food ingredients and by-products"?

The objectives were enumerated and clarified

4. What was the functional idea of the study-system authors had before the beginning of their study and what would be the expected results if that idea was true? Include this as hypothesis and predictions. Having these in the manuscript would improve comprehension of the rationale behind the study.

Additional clarification of the basic concept of the study has bee added before the objectives.

5. The information about food ingredients suppliers (lines 193-213) can be moved to Table 1, indicating with a superscript the supplier of each food ingredient.

This modification was incorporated, and the paragraph was deleted. All the ingredient information is now presented in the footage of Table 1.

6. The materials and methods section should include the rationale for testing the food ingredients considered in the study, and describe how the experimental diets were prepared/mixed.

A sentence was added at the beginning of Materials and Methods to explain the rational of ingredient selection.

Two sentences were added in “Materials and Methods, Phase 2, Diet formulations” to describe the procedure of diet preparation and presentation.

7. I found the statistical analyses a bit confusing. I am struggling to understand which were the explanatory and response variables considered in the experiments and how they were measured, and why mixed models were appropriate for the analysis of such variables. It would be extremely helpful for readers if authors explicitly declare the explanatory and response variables considered in their study indicating their measurement scales, and justify why mixed models, regression, etc. were the appropriate analyses for that specific variables and for reaching the objectives of the study.

Statistical methods have been clarified, variables and units have been specified. Some more information and a reference have been added on the mixed effects GLM, which is better known as generalized linear mixed model GLMM. The only difference is that the GLMM is not restricted to normally distributed data and can be used for binomial data and with skewed data that deviate from normality. 

8. Related to the above, I think readers would be greatly benefited if authors include a graphical representation of the experimental designs in phase 1 and phase 2. Having such a figure would allow readers to better understand how the experimental work was conducted and analyzed.

Although we agree that this study is complex, we disagree with reviewer 1 in that a graphical representation is necessary. We think that the experiment is sufficiently well explained and reviewer 2 did not have any problems understanding it. We respectfully decline adding another figure to a manuscript that already has 8 figures and 10 tables.

9. There are inconsistencies in the use of terminology for referring the same concept, e.g., "total consumed food" (L 275), "total food consumed" (L 276) and "total food consumption" (L 286); or "phase 1 (L 151) and "phase one" (L 153). For sake of clarity, be consistent with the use of terminology throughout the manuscript.

We searched the document and there was only one instance in which “phase one” was used and we corrected it.

We did our best to satisfy reviewer’s 1 request in this case. However, food consumed, and food consumption are two different things. Food consumed is the quantity of food that was consumed, and food consumption is the act of consuming food. In most cases “food consumption” was not changed because it referred to the act of consuming food.

10. In the discussion, why the self-selection approach with crude ingredients have not been used before for developing insect diets (L 895-896)? What are the advantages/disadvantages of the self-selection approach against other approaches for the development and optimization of insect diets (e.g., mixture experiments and response surface methods - see Lapointe et al. 2008, Pascacio-Villafán et al. 2017, Huynh et al. 2019)?

Huynh, M. P., Hibbard, B. E., Lapointe, S. L., Niedz, R. P., French, B. W., Pereira, A. E., ... & Coudron, T. A. (2019). Multidimensional approach to formulating a specialized diet for northern corn rootworm larvae. Scientific reports, 9(1), 3709.

Lapointe, S. L., Evens, T. J., & Niedz, R. P. (2008). Insect diets as mixtures: optimization for a polyphagous weevil. Journal of insect physiology, 54(7), 1157-1167.

Pascacio-Villafán, C., Birke, A., Williams, T., & Aluja, M. (2017). Modeling the costeffectiveness of insect rearing on artificial diets: A test with a tephritid fly used in the sterile insect technique. PloS one, 12(3), e0173205.

A paragraph and three new references have been added to the discussion to address reviewer’s 1 questions.

11. What diet formulation would authors propose as the best in terms of the variables evaluated, and what ingredients do authors propose to investigate further? Please clarify this where appropriate in the abstract and discussion.

A sentence has been added to clarify which of the new diets we consider the best and two sentences have been added to discuss the potential of new ingredients in insect diets.

12. Figure legends are rather vague. For instance, legend for Figure 7 (revise “7A” in line 684) should indicate how to interpret the points below and above the mean, and the points in or out the shaded area.

Figure legends have been corrected

Specific comments:

13. L 69-70, provide examples of expensive commercial feeds with their cost.

We think that the price of commercial feeds can be easily obtained by the reader using the internet and we think it would not be useful to include pricing in the introduction, since that changes rather quickly, and products banish from the market in the regular basis. We respectfully decline to add pricing of products to the introduction.

14. L 99-100, indicate how much canola oil is produced and cite reference.

We think that canola oil production is not relevant as a piece of information for discussion in this paper and reviewers (including reviewer 1) have complain on the size of the introduction. Therefore, we respectfully decline to add information about canola oil production to the introduction.

15. L 115, define oligidic.

A definition was added to the introduction.

16. L 163, containers had a lid? Describe.

Text “without lid” has been added to Materials and Methods.

17. L 164, provide dimensions of the cardboard egg cartons.

This information was added.

18. L 171, insert "crickets" before parenthesis.

Correction done.

19. L 178, for how long oviposition devices were offered to crickets?

Information added.

20. L 190, provide dimensions of the cardboard egg cartons.

Information added.

21. L 257-260, what is "log"? And revise writing.

Corrected.

22. L 273, "... (mg) of ingredient...".

Corrected.

23. L 275, describe how FC was calculated the first time it is mentioned.

Information added.

24. L 287, GLM is more often used for generalized linear models.

The choice of GLMM over GLM is now explained.

25. L 362, Table 2 include five diets, not six.

Corrected.

26. L 452-454, the results of "survival among diet treatments" are found in Figure 7, but where are the results of "cricket survival from first instar to adult"?

Information was added and ANOM procedure results explained in Figure 7.

27. L 453-454, I am unfamiliar to "ANOM", is this correct?

ANOM is correct. Please see Nelson et al. 2005 [45].

28. L 684, 686, 687, include "mean proportion of" as needed. Include error (SE, IC 95%, other) of the mean proportions.

Information added as standard deviations.

29. L 762, "three" not "tree".

Corrected.

30. L 743-745, include Fig or Table showing that specific result.

Tables 3 and 4 cited.

31. L 747, "experiments" or "treatments"? Authors need to be consistent with terminology throughout the manuscript.

Corrected.

32. L 888, the manuscript does not include Figure 9.

Reference to figure 9 deleted.

33. L 895-896, define "crude ingredients".

Definition added.

 Reviewer #2: The article refers to a comprehensive research on the development of

sustainable and convenient diets for the house cricket based on agricultural by-products. The manuscript relies on a well-designed series of experiments, properly arranged and replicated, based on the self-selection of food ingredient, a distinctive procedure which has never been used with Acheta domesticus. In my opinion this is the most interesting aspect for readers, although the findings of this research cover many more aspects, involving diet formulations, economic costs and incomes, impact of macro and micro nutrients. In brief, results are very significant for the discipline. In detail, objectives are clearly enounced as well as the rationale of the research; the methodology, although a bit repetitive, is straightforward and experiments are carefully described. I am not familiar with the specific statistical analyses applied here nor with economic analysis but they sound rigorous. The huge amount of data is presented clearly and supports robust results. The only flaw that can be noticed is the nutritional composition of ingredients which was not obtained through chemical analyses but from literature (as commented at the end of the discussion chapter), anyway results evidence the effectiveness of the applied methodology.

Here below minor misprints are detailed and the respective line number.

Line number Suggestions and comments

253 Amend “depending on” instead of “depending of”

273 Edit “of” instead of “if”

420 Edit “HEPA filter” instead of “HEPHA filter”

550 Insert “within columns” between “Means with the same letter” and “are not significantly

different….”

641 In other figure captions and table titles you specified the unit of the weight, to be

consistent you should specify here “in mg”. Otherwise find a different solution (although mg

appears in the Y-axis title).

683-684 I think there is a misprint: (Fig 7A) does not exist, amend it in “(Fig 7)”

736 I think “for” should be deleted.

816 Edit “latter” instead of “later”

837 Edit “from” instead of “form”

838 Edit “Fusobacterium” instead of “Fusobacterioum”

888 Figure 9 is missing!

897 Amend “relied” instead of “relayed”

All of the above comments have been accepted and corrections have been done.

---

## [Decision Letter · Decision Letter 1]

5 Nov 2019

PONE-D-19-22442R1

Self-Selection of Food Ingredients and Agricultural By-Products by the House Cricket, Acheta domesticus (Orthoptera: Gryllidae): A Holistic Approach to Develop Optimized Diets

PLOS ONE

Dear Dr. Morales-Ramos,

Thank you for submitting your manuscript to PLOS ONE. After careful consideration, we feel that it has merit but does not fully meet PLOS ONE’s publication criteria as it currently stands. Therefore, we invite you to submit a revised version of the manuscript that addresses the points raised during the review process.

The current version of the manuscript has much improved, however there are some details that still need to be addressed. Please carefully consider the reviewer's comments and recommendations.

We would appreciate receiving your revised manuscript by Dec 20 2019 11:59PM. To enhance the reproducibility of your results, we recommend that if applicable you deposit your laboratory protocols in protocols.io, where a protocol can be assigned its own identifier (DOI) such that it can be cited independently in the future. For instructions see: http://journals.plos.org/plosone/s/submission-guidelines#loc-laboratory-protocols

We look forward to receiving your revised manuscript.

Kind regards,

Nicoletta Righini, PhD

Academic Editor

PLOS ONE

Additional Editor Comments (if provided):

I agree with the reviewer that Fig. 2 should go into the Supplementary material (By the way, re: reviewer´s comment #4, on Fig. 2 I see the egg-cartons placed horizontally, not vertically).

Line 129 - ...for a diet that convergeS (an 's' is missing)

Line 445-446: I find the phrasing '..using THE generalized linear mixed model' odd, since there is not only one GLMM (i.e., you should say 'a' GLMM). I also agree with the reviewer that the Methods would be much clearer if you could organize better the variables and models you ran (e.g., Model 1: response variable xx, random factor xx, fixed factors xx; Model 2: xxx, etc...)

Line 453: you still mention 'mixed effects GLM'

Reviewers' comments:

Reviewer's Responses to Questions

**Comments to the Author**

1. If the authors have adequately addressed your comments raised in a previous round of review and you feel that this manuscript is now acceptable for publication, you may indicate that here to bypass the “Comments to the Author” section, enter your conflict of interest statement in the “Confidential to Editor” section, and submit your "Accept" recommendation.

Reviewer #1: (No Response)

2. Is the manuscript technically sound, and do the data support the conclusions?

Reviewer #1: Yes

3. Has the statistical analysis been performed appropriately and rigorously? 

Reviewer #1: Yes

4. Have the authors made all data underlying the findings in their manuscript fully available?

Reviewer #1: No

5. Is the manuscript presented in an intelligible fashion and written in standard English?

Reviewer #1: Yes

6. Review Comments to the Author

Reviewer #1: Thank you to the authors for their revision of manuscript PONE-D-19-22442R1.

1. I must say it was a bit difficult to assess the revision of this manuscript because in their response to reviewers’ letter, authors did not indicate the line number of the changes in the manuscript. Please always include line numbers in your response to reviewers’ letter. Otherwise, you make the work of reviewers unnecessarily difficult. There are also inconsistencies between the response to reviewers’ letter and the actions taken in the manuscript. For instance, response to Specific comment # 13 reads "... We respectfully decline to add pricing of products in the introduction"; however, prices were actually included in the introduction (L 72-73). This situation made me feel confused.

2. The manuscript still has sections that should improve in precision and clarity. Most of them related to the experimental design and statistical analyses. In this regard, I think authors misunderstood my comment # 8, as I did not say it was "necessary" to include a graphical representation of the experimental designs. What I say, was that "readers would be greatly benefited" if such a figure was included. I have no objection if authors refuse to follow this comment. However, consider that many readers will find it much easier to understand complex studies with a graphical explanation. But again, this is not mandatory, but rather something that could add to improve the comprehension of the study.

3. What I must insist, is that the response and explanatory variables considered in experiments and analyses must be clearly mentioned in the materials and methods section. For instance, it is not completely clear how the "Data consisting of ending biomass and total food consumed" (L 279) was measured and its units. Also, in the results section it is mentioned that "Sex had a significant impact on adult dry weight..." (L 667-668), but in the materials and methods section it is never mentioned that sex was considered as an explanatory variable in the models fitted to data. Authors must link all the pieces of information in the materials & methods with the results to improve the flow of ideas and the comprehension of their study.

4. Throughout the materials and methods section, it is mentioned that egg cartons were placed horizontally, but  in the figure 2 it seems to me that the cartons were placed in vertical position.

5. Correct kilogram symbol throughout the manuscript (kg - correct; Kg - incorrect).

6. L 42, the remaining 38 and 8% of new diet formulations consisted of...

7. L 67-68, "... primitive rearing practices that require..." this is not necessarily true. There are companies using state-of-the-art technologies to rear crickets. For instance, see ASPIRE food group.

8. L 92-94, what nutrients, exactly, are found in by-products, and cite reference.

9. L 151, indicate conditions of egg hatch.

10. L 153, indicate long, wide and tall. Applies to all instances in which size is indicated.

11. L 199-202, is there another way in which authors can present this information? In its current form it is confusing.

12. In Table 1, please include the amount of each food that was offered to crickets.

13. L 232-236, I think this figure indicate proportions not ratios.

14. In Table 2, food ingredients of diet 2 do not sum to 100; please correct. Suggest "Estimated nutrient proportions" instead of "Estimated nutrient ratios".

15. L 333-334, indicate how much food was added to Petri dishes.

16. L 339, remove extraneous period.

17. L 381-382, remove "The live weight...", this is already mentioned at the end of last paragraph.

18. L 428-429, how much food was provided to crickets.

19. Line 505 - Table 3, indicate the period in which these foods were consumed, and the number of crickets that consumed this amount of food.

20. I am wondering if some of the tables or figures (perhaps figure 2) could be moved to supplementary material.

7. PLOS authors have the option to publish the peer review history of their article (what does this mean?). If published, this will include your full peer review and any attached files.

Reviewer #1: No

---

## [Author Response · Author response to Decision Letter 1]

12 Dec 2019

Answer to reviewers:

Editor Comments:

Thank you for submitting your manuscript to PLOS ONE. After careful consideration, we feel that it has merit but does not fully meet PLOS ONE’s publication criteria as it currently stands. Therefore, we invite you to submit a revised version of the manuscript that addresses the points raised during the review process.

The current version of the manuscript has much improved, however there are some details that still need to be addressed. Please carefully consider the reviewer's comments and recommendations.

Additional Editor Comments (if provided):

I agree with the reviewer that Fig. 2 should go into the Supplementary material (By the way, re: reviewer´s comment #4, on Fig. 2 I see the egg-cartons placed horizontally, not vertically).

Figure 2 has been moved to supplementary materials and all the rest of the figures have been renumbered.

Indeed, the stack of cartons were placed horizontally. Reviewer 1 may have been thinking about individual cartoon pieces on the whole stack.

Line 129 - ...for a diet that convergeS (an 's' is missing)

Corrected.

Line 445-446: I find the phrasing '..using THE generalized linear mixed model' odd, since there is not only one GLMM (i.e., you should say 'a' GLMM). I also agree with the reviewer that the Methods would be much clearer if you could organize better the variables and models you ran (e.g., Model 1: response variable xx, random factor xx, fixed factors xx; Model 2: xxx, etc...)

The description of GLMM has been place between parentheses instead of GLMM.

Description of the models have been added to the text.

Line 453: you still mention 'mixed effects GLM'

Corrected.

Reviewer #1: Thank you to the authors for their revision of manuscript PONE-D-19-22442R1.

1. I must say it was a bit difficult to assess the revision of this manuscript because in their response to reviewers’ letter, authors did not indicate the line number of the changes in the manuscript. Please always include line numbers in your response to reviewers’ letter. Otherwise, you make the work of reviewers unnecessarily difficult. There are also inconsistencies between the response to reviewers’ letter and the actions taken in the manuscript. For instance, response to Specific comment # 13 reads "... We respectfully decline to add pricing of products in the introduction"; however, prices were actually included in the introduction (L 72-73). This situation made me feel confused.

We did not indicate the line numbers where changes were done because we were under the impression that reviewers had access to the “track changes” version of the manuscript. We will provide line numbers this time. We apologize for the inconsistencies, but clarity is also needed from the reviewer’s side. Some of the comments were difficult to understand and we were just trying to address them the best we could. We will try to do better this time.

2. The manuscript still has sections that should improve in precision and clarity. Most of them related to the experimental design and statistical analyses. In this regard, I think authors misunderstood my comment # 8, as I did not say it was "necessary" to include a graphical representation of the experimental designs. What I say, was that "readers would be greatly benefited" if such a figure was included. I have no objection if authors refuse to follow this comment. However, consider that many readers will find it much easier to understand complex studies with a graphical explanation. But again, this is not mandatory, but rather something that could add to improve the comprehension of the study.

The way we interpreted some of the comments from reviewer 1 may have not been as intended. We are also confused by difficulties of the reviewer to understand the statistical procedures, which are not outside of the standard for this type of studies. Reviewer 2 from the first submission, did not have such difficulties.

3. What I must insist, is that the response and explanatory variables considered in experiments and analyses must be clearly mentioned in the materials and methods section. For instance, it is not completely clear how the "Data consisting of ending biomass and total food consumed" (L 279) was measured and its units. Also, in the results section it is mentioned that "Sex had a significant impact on adult dry weight..." (L 667-668), but in the materials and methods section it is never mentioned that sex was considered as an explanatory variable in the models fitted to data. Authors must link all the pieces of information in the materials & methods with the results to improve the flow of ideas and the comprehension of their study.

A sentence was missing explaining that crickets were weighed at the end of the experiment. This was added (Lines 254-256). 

How food consumption was calculated is well explained in the first paragraph in “Data Analysis”. Explanation of how the data was collected is explained in the previous paragraph before the section “Data Analysis” (lines 258-264).

Please see these: “After six weeks, experimental units were monitored daily for the presence of adult crickets. When present, adult crickets were sexed, weighed, and frozen at -25ºC. The live weight and sex of each cricket was recorded along with its corresponding experimental unit number, diet treatment and date of emergence.” Lines (426-429).

4. Throughout the materials and methods section, it is mentioned that egg cartons were placed horizontally, but in the figure 2 it seems to me that the cartons were placed in vertical position.

Horizontally is correct

5. Correct kilogram symbol throughout the manuscript (kg - correct; Kg - incorrect).

This has been corrected throughout the text.

6. L 42, the remaining 38 and 8% of new diet formulations consisted of...

This is incorrect. The percentages as stated in the text are correct.

7. L 67-68, "... primitive rearing practices that require..." this is not necessarily true. There are companies using state-of-the-art technologies to rear crickets. For instance, see ASPIRE food group.

Some companies, unknowingly to our team, may have been achieving progress in this area, but they are a very small exception of the commercial companies world-wide. Still, we submit that “state of the art” may still be primitive as compared to livestock industry.

8. L 92-94, what nutrients, exactly, are found in by-products, and cite reference.

References have been provided and (lines 206-207). Enumerating the nutrients of each by-product will require additional tables, but this is unnecessary because the literature provided has this information in downloadable form.

9. L 151, indicate conditions of egg hatch.

Done (Line 152).

10. L 153, indicate long, wide and tall. Applies to all instances in which size is indicated.

Done (Lines 153, 166, 171, 181, and 407).

11. L 199-202, is there another way in which authors can present this information? In its

current form it is confusing.

This way is no confusing is just a simple mathematical operation.

12. In Table 1, please include the amount of each food that was offered to crickets.

Done.

13. L 232-236, I think this figure indicate proportions not ratios.

Ratios is correct, but they are in three axes instead of two.

14. In Table 2, food ingredients of diet 2 do not sum to 100; please correct. Suggest

"Estimated nutrient proportions" instead of "Estimated nutrient ratios".

Corrected (Thank you for this one).

15. L 333-334, indicate how much food was added to Petri dishes.

Done (Lines 328-330).

16. L 339, remove extraneous period.

Done (thank you).

17. L 381-382, remove "The live weight...", this is already mentioned at the end of last

paragraph.

Corrected (Lines 376-377).

18. L 428-429, how much food was provided to crickets.

This information was added (Line 415).

19. Line 505 - Table 3, indicate the period in which these foods were consumed, and the number of crickets that consumed this amount of food.

This is well explained in “Materials and Methods” (Lines 179-180 and 239-243). Experiment duration and initial cricket population per experimental unit. Crickets were not counted at the end because this information was no relevant for the self-selection study. Relative ingredient consumption was the relevant information required. 

20. I am wondering if some of the tables or figures (perhaps figure 2) could be moved to

supplementary material.

To be able to satisfy this, we need to know specifically what tables and an explanation of why it would be better to move them to supplementary material. The tables are currently presenting information that it would be difficult to present in any other way.

---

## [Editor Report · Decision Letter 2]

19 Dec 2019

Self-Selection of Food Ingredients and Agricultural By-Products by the House Cricket, Acheta domesticus (Orthoptera: Gryllidae): A Holistic Approach to Develop Optimized Diets

PONE-D-19-22442R2

Dear Dr. Morales-Ramos,

We are pleased to inform you that your manuscript has been judged scientifically suitable for publication and will be formally accepted for publication once it complies with all outstanding technical requirements.

With kind regards,

Nicoletta Righini, PhD

Academic Editor

PLOS ONE
---

## [Editor Report · Acceptance letter]

3 Jan 2020

PONE-D-19-22442R2 

Self-Selection of Food Ingredients and Agricultural By-Products by the House Cricket, *Acheta domesticus* (Orthoptera: Gryllidae): A Holistic Approach to Develop Optimized Diets 

Dear Dr. Morales-Ramos:

I am pleased to inform you that your manuscript has been deemed suitable for publication in PLOS ONE. Congratulations! Your manuscript is now with our production department. 

With kind regards,

on behalf of

Dr. Nicoletta Righini 

Academic Editor

PLOS ONE